# Mapping protein states and interactions across the tree of life with co-fractionation mass spectrometry

Michael A. Skinnider [1,2,3], Mopelola O. Akinlaja [1,4] & Leonard J. Foster [1,4] ✉

We present CFdb, a harmonized resource of interaction proteomics data from 411 co-fractionation mass spectrometry (CF-MS) datasets spanning 21,703 fractions. Meta-analysis of this resource charts protein abundance, phosphorylation, and interactions throughout the tree of life, including a reference map of the human interactome. We show how large-scale CF-MS data can enhance analyses of individual CF-MS datasets, and exemplify this strategy by mapping the honey bee interactome.

Cellular processes arise from the dynamic organization of proteins in networks of physical interactions. Significant resources have been devoted to mapping the protein interaction networks of humans and model organisms[1]. These networks are widely used for tasks such as protein function prediction, disease gene prioritization, or interpretation of transcriptomic and proteomic datasets[2,3]. However, questions about the reproducibility of these networks have persisted. Limited overlap between screens performed in different laboratories was noted soon after the first maps of the yeast interactome emerged[4,5]. Two decades later, large-scale efforts have produced systematic maps of the human interactome that display relatively little overlap with one another, with a mean Jaccard index of just 0.062 between any pair of networks (Supplementary Fig. 1). This lack of overlap has been variously attributed to differences in the types of interactions detected by each assay and the proteins targeted by individual screens, variation in experimental protocols or the depth of protein identification, the presence of context-specific interactions, or false-positives and false-negatives in the resulting interactome maps.

CF-MS has emerged as a powerful technique to map protein interaction networks, particularly under physiological conditions and in non-model organisms[6,7]. However, CF-MS datasets are generally collected within individual laboratories and analyzed in isolation. At the same time, the increasingly wide uptake of CF-MS has led to hundreds of datasets being deposited in public proteomic databases. This wealth of data opens up opportunities for larger-scale data integration to reveal patterns that are reproducible across dozens or even hundreds of datasets.

We previously described a meta-analysis of 206 uniformly processed CF-MS datasets, and used this data to establish best practices for the design and analysis of CF-MS experiments[8]. Here, we double the size of this resource by re-analyzing a further 205 experiments using the same pipeline (Supplementary Data 1 and Supplementary Fig. 2a, b). The updated resource, which we named CFdb, now comprises 20.1 million measurements of protein abundance derived from 128.7 million sequenced peptides across 21,703 fractions. Proteomic analysis of all 21,703 fractions required a total of 43.2 months of uninterrupted instrument time (Fig. 1a, b), emphasizing the value of meta-analysis to assemble a resource whose scale would make it impractical to acquire within a single laboratory.

## Results

CFdb incorporates data for eight species that were not represented in our original dataset, but also reflects major expansions in the data available for human, mouse, *Arabidopsis*, and yeast (Supplementary Fig. 2c, d). For example, reanalysis of an additional 120 datasets increased the amount of human CF-MS data by a factor of 2.5×, enabling the detection of 1958 human proteins that were not represented in the original resource and markedly increasing the number of well-quantified proteins (e.g., those detected in at least 500 fractions; Fig. 1c, Supplementary Fig. 3a, b, and Supplementary Data 2). Similar expansions in proteome coverage were observed for mouse, yeast, and *Arabidopsis* (Supplementary Fig. 3c–k). Proteins that were detected exclusively in the updated resource tended to be of low abundance and tissue-specific (Fig. 1d, e and Supplementary Fig. 4a–e). Saturation analysis suggested that, at least for human, additional CF-MS data

[1]Michael Smith Laboratories, University of British Columbia, Vancouver, BC, Canada. [2]Lewis-Sigler Institute for Integrative Genomics, Princeton University, Princeton, NJ, USA. [3]Ludwig Institute for Cancer Research, Princeton University, Princeton, NJ, USA. [4]Department of Biochemistry and Molecular Biology, University of British Columbia, Vancouver, BC, Canada. ✉e-mail: foster@msl.ubc.ca

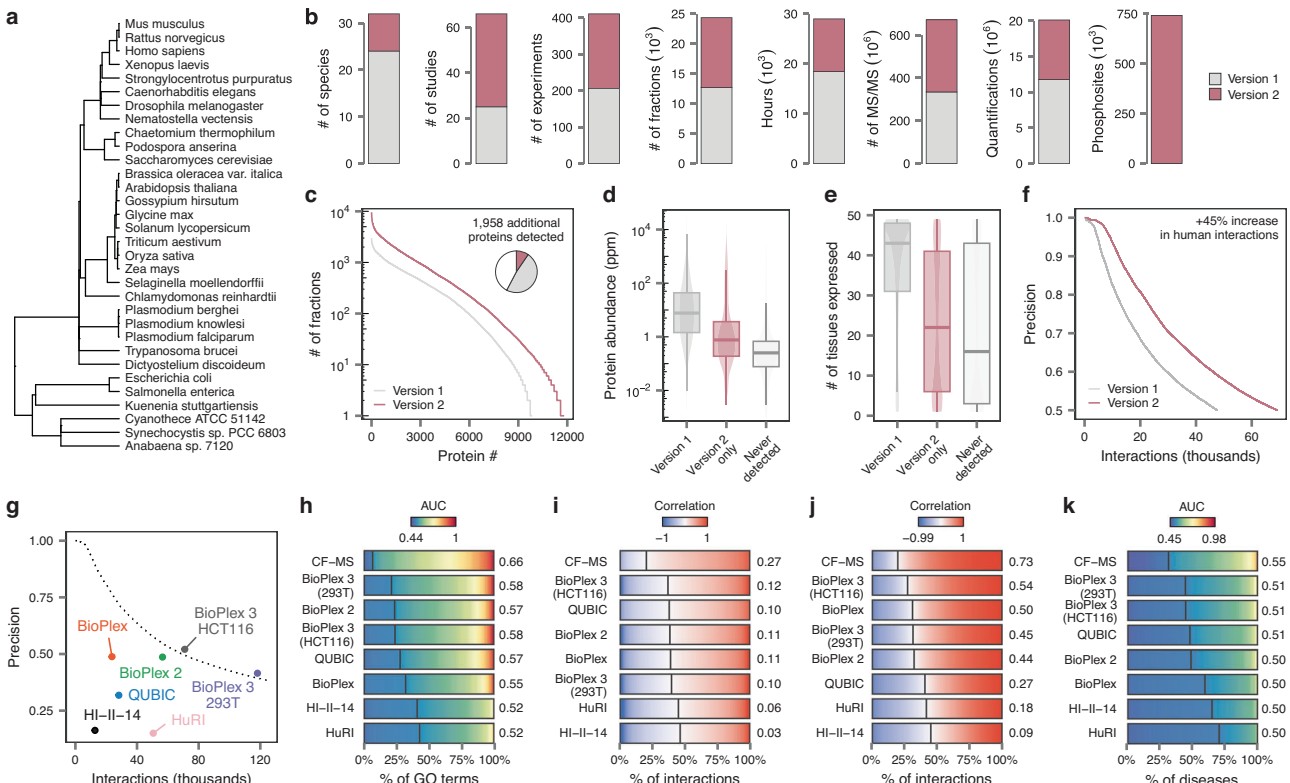

**Fig. 1 | A harmonized resource of CF-MS data charts protein abundance and interactions across the tree of life. a** Phylogenetic tree showing the 32 species with CF-MS data included in CFdb. **b** Expansions to the scope and coverage of CF-MS data in CFdb ("version 2"), as compared to our original meta-analysis ("version 1"). Phosphosite quantifications are assigned exclusively to version 2 because CF-MS datasets were not searched for phosphopeptides in our original meta-analysis. **c** Cumulative distribution function showing the number of fractions in which each human protein was quantified. Inset pie chart shows the total proportion of human proteins detected in at least one CF-MS fraction. **d** Abundance of human proteins detected by CF-MS in the original meta-analysis or the updated resource, versus those never detected by CF-MS, based on consensus protein abundance estimates from the PaxDb database[61] (for $n = 8{,}248$ proteins overlapping between CFdb and PaxDb). **e** Tissue specificity of human proteins detected by CF-MS in the original meta-analysis or the updated resource, versus those never detected by CF-MS (for $n = 10{,}978$ proteins overlapping between CFdb and the Human Protein Atlas).

**f** Precision of the human interactome inferred by meta-analysis of CF-MS experiments in CFdb as compared to our original meta-analysis, for interaction networks of a given size. **g** Precision of the human interactome inferred by meta-analysis of CF-MS experiments in CFdb for interaction networks of a given size, as compared to six high-throughput screens of the human interactome using Y2H or AP-MS. **h–k**, Comparisons to previous interactome screens highlight the quality of the CFdb human interactome. **h** Functional coherence of interactome networks, as quantified by the AUC of protein function prediction in cross-validation. Text shows the median AUC. Vertical lines show the proportion of GO terms with AUC less than 0.5, equivalent to random chance. **i** Coexpression of interacting protein pairs across a large proteomic dataset. Text shows the median Pearson correlation. Vertical lines show the proportion of negatively correlated pairs[88]. **j** Colocalization of interacting protein pairs by subcellular proteomics. **k** Connectivity between genes associated with the same disease, as quantified by the AUC of disease gene prediction in cross-validation. Source data are provided as a Source Data file.

would bring diminishing returns: we estimated that doubling the number of human CF-MS experiments again would increase the number of quantified proteins by just ~1200, from 11,912 to 13,155 (Supplementary Fig. 4f, g).

We used CFTK[8] to reconstruct protein interaction networks for every species in CFdb profiled in at least three experiments, including several not represented in our original resource. For example, meta-analysis of five CF-MS experiments in *E. coli* allowed us to reconstruct a network of 5276 protein-protein interactions (Supplementary Fig. 5a). Similarly, the inclusion of seven additional yeast experiments allowed us to construct a draft map of the yeast interactome by CFMS (Supplementary Fig. 5b).

The expanded dataset also afforded marked improvements for previously reported networks. In human, the inclusion of 120 additional CF-MS experiments allowed us to recover an additional 21,545 interactions, an increase of 45% (Fig. 1e, Supplementary Fig. 5c and Supplementary Data 3). Moreover, despite this increase in size, multiple lines of evidence suggested that the updated human interactome was of comparable or higher quality to our original draft. Proteins implicated in the same biological processes showed a similar propensity to interact with one another in either network, and interacting

proteins showed similar patterns of correlation across large proteomic datasets or colocalization to the same subcellular compartments (Supplementary Fig. 5d–h). Interestingly, interactions found only in the updated network were significantly more likely to overlap with at least one small-scale or high-throughput experiment, supporting the notion that meta-analysis can improve the reproducibility of biological networks ($p = 1.9 \times 10^{-15}$, $\chi^2$ test; Supplementary Fig. 5i–j), although the overlap between the CF-MS interactome and previous high-throughput screens remained relatively low (Supplementary Fig. 5k). Likewise, the addition of 19 mouse CF-MS experiments allowed us to recover an additional 19,251 interactions, and multiple lines of evidence suggested that the updated mouse interactome was of higher quality than the networks we had previously reported for seven mouse tissues[9] (Supplementary Fig. 6).

To place the human CF-MS interactome in context, we compared it to seven large-scale interactome screens using affinity purification-mass spectrometry (AP-MS) or yeast two-hybrid (Y2H). Evaluating these screens on identical sets of true positive and true negative interactions derived from the CORUM database of protein complexes[10], we found the CF-MS interactome was of comparable or higher quality to any of these screens at equivalent precision (Fig. 1f).

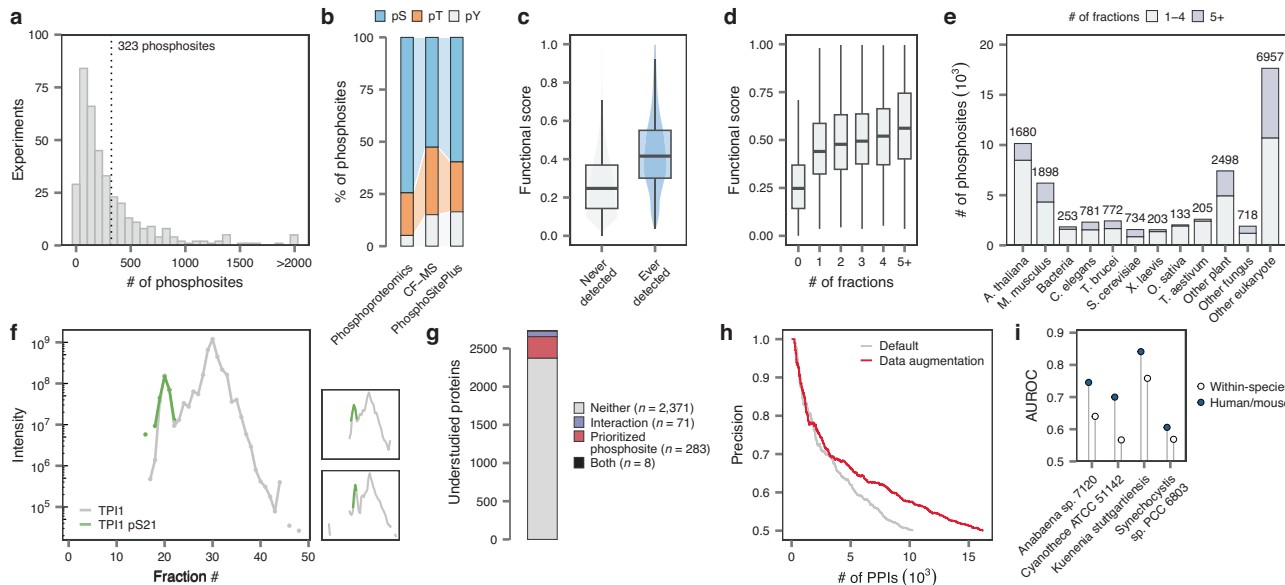

**Fig. 2 | CFdb prioritizes functional phosphosites and enhances analysis of CF-MS data from non-model organisms. a** Histogram showing the number of phosphoproteins quantified in each CF-MS experiment. **b** Proportion of phosphoserine (pS), phosphothreonine (pT) and phosphotyrosine (pY) residues in a large-scale meta-analysis of the human phosphoproteome, as compared to all phosphosites detected by CF-MS or phosphosites from the curated PhosphoSite-Plus database[16]. **c** Functional scores of human phosphosites that were or were not ever detected by CF-MS ($n = 116{,}258$ phosphosites). **d** Functional scores of human phosphosites, stratified by the number of CF-MS fractions in which each phosphosite was detected ($n = 116{,}258$ phosphosites). **e** Number of phosphosites prioritized based on detection in at least five fractions across major non-human species or taxonomic groups in CFdb. **f** Example of a frequently quantified phosphosite, pS21 of TPI1. Chromatograms show the intensity of pS21-containing phosphopeptides (green) or the parent protein (light gray). **g** Overview of understudied human proteins for which an interaction or prioritized phosphosite was detected in CFdb. **h** Precision of the honey bee CF-MS interactome, inferred with and without a data augmentation strategy that leverages data from hundreds of experiments in CFdb. **i** Separation of intra- and intra-complex interactions in interactome networks reconstructed for four prokaryotes by random forest classifiers trained in cross-validation on species-specific protein complexes ("within-species") versus on 206 human and mouse CF-MS experiments ("human/mouse"), as quantified by the area under the ROC curve (AUROC). Source data are provided as a Source Data file.

This conclusion was further corroborated by the functional coherence, coexpression, and colocalization of interacting proteins within each network (Fig. 1g–i and Supplementary Fig. 5l–m). Intriguingly, genes associated with the same disease showed a particularly strong tendency to interact with one another in the CF-MS interactome as compared to other screens, suggesting this network could be particularly valuable in interpreting exome sequencing or genome-wide association studies[11,12] (Fig. 1k).

CF-MS has been used to identify phosphorylation-dependent interactions by comparing phosphorylase-treated and untreated samples[13]. We asked whether large-scale CF-MS data, collected without bespoke experimental strategies, could inform on the phosphorylation state of interacting proteins. Searching all 411 experiments for phosphopeptides yielded a resource of 742,040 phosphosite quantifications (Fig. 1b). Despite the lack of phosphopeptide enrichment, an average of 323 phosphosites and 183 phosphoproteins were quantified per experiment, and many individual phosphosites were measured across hundreds of fractions (Fig. 2a and Supplementary Fig. 7a–h). Phosphosites detected in a greater number of fractions or experiments tended to be more technically reliable (as quantified by either the localization probability or the delta score) and display a higher stoichiometry to the unmodified peptide (Supplementary Fig. 7i–p). Interestingly, whereas radioisotope[14] and phosphoproteomic[15] data suggest that phosphotyrosines (pY) account for less than 1% of the phosphoproteome, we observed a substantially greater proportion of pY sites in CF-MS data (Fig. 2b and Supplementary Fig. 8a–c). Tyrosine residues accounted for 15.1% of the phosphosites detected by CF-MS, similar to the proportion in the curated PhosphoSitePlus database[16] (16.5%) but greater than that observed in the largest meta-analysis of phospho-enriched proteomics datasets to date (5.1%)[17]. Given that most CF-MS protocols enrich for protein complexes by design, this

observation raised the possibility that tyrosine phosphosites might be overrepresented within protein complexes. In agreement with this possibility, pY-containing proteins were also enriched among protein complexes in a large-scale meta-analysis of phosphoproteomic datasets[17] (Supplementary Fig. 8d), and this enrichment remained statistically significant when controlling for protein abundance ($p = 9.4 \times 10^{-4}$, logistic regression).

Phosphoproteomic experiments typically detect thousands of phosphosites, but assigning function to these sites is challenging[17,18]. One promising approach involves using machine learning to integrate multiple sources of information into a single score reflecting the functional relevance of a given phosphosite[17]. Remarkably, we found that phosphosites that were detected by CF-MS had significantly higher functional scores than the proteome average (Fig. 2c). Moreover, phosphosite functional scores increased with the number of CF-MS fractions in which a phosphosite was detected, to the point that phosphosites detected in five or more fractions had functional scores similar to those of known regulatory phosphosites[17] (Fig. 2d). Some of these phosphosites may play a role in regulating protein-protein interactions. For example, one of the most frequently quantified human phosphosites was S21 of TPI1, a residue whose phosphorylation was recently reported to regulate the assembly of this protein into homodimers[19]. Consistent with this notion, S21 phosphopeptides were specific to one of two chromatographic peaks across CF-MS datasets (Fig. 2f). Similarly, we identified multiple frequently quantified phosphosites in NES, but found that whereas most of these were detected in the highest-intensity CF-MS peak, peptides phosphorylated at S1347 were specific to a secondary, low-intensity peak (Supplementary Fig. 7q).

Phosphosites detected in five or more CF-MS fractions had a number of unusual features relative to the phosphoproteome average.

In an independent phosphoproteomic dataset, these phosphosites tended to be identified in more samples, by more MS/MS spectra, and with higher localization probabilities, all features characteristic of more reliable identifications (Supplementary Fig. 9a–c). They were also more likely to be located in intrinsically disordered regions[20] and in recurrently phosphorylated structural regions (phosphorylation 'hotspots'[21]), both of which are known to be enriched for functional phosphosites (Supplementary Fig. 9d–e). Frequently quantified phosphosites were more likely to match known protein kinase motifs, and kinase-substrate enrichment analysis[16] identified several kinases whose known substrates were enriched among these phosphosites, with the most significant enrichments for CK2A1, CDK1, and CDK2 (Supplementary Fig. 9f–h). Last, estimates of phosphosite evolutionary age[22] suggested that frequently quantified phosphosites were evolutionarily ancient, with human-specific phosphosites depleted and phosphosites conserved across tetrapods or bilaterians enriched among this set (Supplementary Fig. 9i).

Calculation of phosphosite functional scores requires the laborious integration of proteomic, structural, regulatory and evolutionary datasets, which has so far limited this approach to the human proteome. The observation that phosphosites quantified in at least five CF-MS fractions display functional scores comparable to known regulatory phosphosites suggested that detection by CF-MS could be used to prioritize functional phosphosites across species. To evaluate this possibility, we searched for phosphosites detected in at least five CF-MS fractions in other species, recognizing that imposed a somewhat more stringent criterion because fewer fractions were profiled by CF-MS in species other than human (Supplementary Fig. 9j). Beyond human, an average of 543 phosphosites per species met this criterion, yielding a total of 16,832 phosphosites across 31 species (Fig. 2e and Supplementary Data 4). Together, these results suggest that CFdb can contribute to prioritization of functional phosphosites across species, although it is important to emphasize that CF-MS does in and of itself not provide direct evidence of function.

Proteome-scale technologies like CF-MS have attracted interest for their potential to shed light on the functions of poorly characterized proteins[23]. Among the 11,912 human proteins quantified by CF-MS, 488 were linked to 10 or fewer PubMed IDs. Of these, our meta-analysis detected a protein-protein interaction or prioritized a regulatory phosphosite for 346, supporting the potential for large-scale data integration approaches to assist in functional annotation (Fig. 2g and Supplementary Fig. 10).

To date, the dominant paradigm in the field has been to analyze each newly collected CF-MS dataset in isolation. We asked whether CFdb could augment analyses of smaller-scale CF-MS projects, such as those carried out within individual laboratories. We reasoned that such approaches could be particularly useful for studies of non-model organisms, in which few protein complexes may be known.

As a proof of concept, we carried out a new set of CF-MS experiments in honey bee (*Apis mellifera*). Honey bees are pollinators that play central roles in global agriculture[24,25]. However, the honey bee interactome remains largely unmapped. This gap reflects a number of challenges that are common to the study of non-model organisms, including a lack of established cell culture systems, limited amenability to genetic manipulation, and incomplete proteome annotation. CF-MS is well-suited to overcome these limitations by enabling interactome mapping under physiological conditions within in vivo tissues, and without requiring validated antibodies or the introduction of protein tags.

As a first step towards mapping the honey bee interactome, we profiled the honey bee midgut by CF-MS. The midgut was selected as the primary site of infection for a prevalent honey bee pathogen, *Vairimorpha (Nosema) ceranae*, that has been implicated in the collapse of honey bee colonies. Ten CF-MS experiments were performed in which 40 fractions were collected and mass spectrometry data was acquired with data-independent acquisition (DIA), while an eleventh experiment was analyzed using data-dependent acquisition (DDA). An average of 2092 proteins were quantified per replicate, yielding a total of 319,105 protein quantifications across all 440 fractions.

We then sought to infer protein interaction networks for the honey bee midgut from this dataset. However, of the 5163 human proteins within the CORUM database[10], only 1200 could be mapped to a bee ortholog, limiting the amount of training data available for network inference. We therefore devised a data augmentation strategy that leveraged CFdb to augment our bee data with labeled protein pairs that were sampled randomly from 206 human or mouse CF-MS experiments (Supplementary Fig. 11a). This strategy increased the size of the honey bee CF-MS interactome by 64.7%, while simultaneously improving its functional coherence (Fig. 2h, Supplementary Fig. 11b, c, and Supplementary Data 5). Moreover, interacting proteins in either network showed comparable patterns of coexpression across a large honey bee proteomic dataset[26], and their fly orthologs displayed comparable patterns of co-elution in an independent CF-MS dataset[27] (Supplementary Fig. 11d, e). Data augmentation enabled improved coverage of several CORUM protein complexes with one-to-one orthologs in honey bee, such as the 26 S proteasome, the LSm2-8 complex, and the 20 S methylosome (Supplementary Fig. 11f). Beyond better coverage of known complexes, data augmentation also enabled the identification of previously unidentified interactions. For instance, we identified an interaction between calreticulin and inositol3-phosphate synthase. This interaction was not annotated in any species in the BioGRID database[28], but these proteins were previously found to co-purify in a large AP-MS study of the fly interactome[29], supporting the existence of an orthologous interaction in honey bee (Supplementary Fig. 11g, h).

We also asked whether CFdb could enable interactome mapping without requiring a training dataset of known protein complexes. To explore this possibility, we focused on four understudied prokaryotes (*Anabaena* sp. 7120, *Cyanothece* sp. ATCC 51142, *Kuenenia stuttgartiensis*, and *Synechocystis* sp. PCC 6803) represented in CFdb[30–33]. Among the known protein complexes in the EcoCyc database[34], just 158 to 190 intra-complex interactions could be mapped to one-to-one orthologs in each of these four species, presenting a challenge to the supervised machine-learning approach that is the dominant paradigm in CF-MS data analysis. We hypothesized that a machine-learning model trained on aggregate patterns of protein complex co-elution across 206 human or mouse CF-MS experiments could enable network inference in these species without requiring a training set of species-specific protein complexes. Consistent with this hypothesis, a random forest classifier trained on human and mouse experiments better separated intra- and inter-complex interactions derived from EcoCyc than supervised machine-learning within each species (Fig. 2i and Supplementary Fig. 12a). Moreover, protein interaction networks derived from the classifier trained on human and mouse experiments also demonstrated a higher degree of functional coherence than networks derived from within-species machine-learning (Supplementary Fig. 12b). These findings indicate that for non-model organisms in which few known protein complexes are available to train a classifier, learning from human and mouse CF-MS datasets can enable de novo network inference.

## Discussion

The increasing uptake of CF-MS has led to the deposition of hundreds of datasets in public proteomic repositories. However, these datasets are generally collected and analyzed in isolation by individual laboratories. Here, we explored the possibility of aggregating biologically and technically heterogeneous CF-MS data at the repository scale. We reanalyzed 21,703 fractions from 411 CF-MS experiments using a uniform computational pipeline that standardized protein identification, quantification, and quality control. This expanded resource

incorporated data from eight additional species and substantially expanded the proteome coverage of humans and model organisms by CF-MS. It also dramatically improved our ability to infer protein interaction networks through meta-analysis of all available CF-MS data for any given species. For example, through meta-analysis of 166 CF-MS experiments, we have produced a map of the human interactome that multiple lines of evidence suggest is among the highest-quality interactome maps currently in existence. Similarly, we present a high-quality map of the mouse interactome derived from meta-analysis of 40 CF-MS experiments, as well as CF-MS interactomes for major model organisms not covered in our original resource such as yeast or *E. coli*. These networks include interactions for hundreds of low-abundance, tissue-specific, and/or understudied proteins not captured in our original resource. We carried out extensive comparisons to other functional genomics datasets that substantiate the quality of the inferred networks at a systems level. However, we caution users of this resource that any particular interaction should in isolation be regarded as putative until confirmed experimentally using an orthogonal technique.

Beyond protein-protein interactions, we also carried out a meta-analysis of protein phosphorylation across all 21,703 fractions. This resource adds to the relatively small number of phosphoproteomic meta-analyses that have been carried out to date[17,35,36], and provides a dataset that can be used to answer a number of different biological questions or contribute to the development of new computational tools, such as methods to computationally predict phosphorylation sites[37,38] or phosphopeptide mass spectra[39–41]. This resource is also unique in that protein phosphorylation has rarely been investigated with CF-MS, except by studies that employed bespoke experimental designs[13]. Here, we show that, despite the lack of explicit phosphopeptide enrichment in published CF-MS datasets, many phosphosites are detected across dozens or hundreds of fractions. These frequently detected phosphosites have a number of unusual features relative to the phosphoproteome average. They are evolutionarily ancient, enriched in intrinsically disordered regions and phosphorylation hotspots, more likely to match known protein kinase motifs, and enriched for the targets of several protein kinases. These properties are reflected in the assignment of disproportionately high functional scores to frequently-detected phosphosites by one machine-learning approach[17], relative to both phosphosites ever detected by CF-MS and to the human phosphoproteome in general. Together, these observations suggest that CFdb can contribute a new source of data to efforts to prioritize functional phosphosites across species. However, two major limitations of our analysis are important to note: CF-MS does not in and of itself provide direct evidence of phosphosite function, and here we used a simple heuristic of detection in at least five fractions to prioritize phosphosites based on CF-MS. Although this heuristic allowed us to study a number of interesting features of frequently detected phosphosites, integrating data from CFdb with other proteomic, structural, regulatory and evolutionary datasets will provide a more robust basis for phosphosite prioritization in the future.

Our meta-analytical approach has both strengths and limitations. Our goal in this study was to leverage the vast quantities of CF-MS data that have been deposited to public repositories in order to identify protein interactions supported by evidence from many biologically and technically heterogeneous experiments. In other words, we sought to identify protein pairs that consistently demonstrated co-elution across different cell lines or tissues, experimental protocols, and mass spectrometric methods, with the expectation that these pairs would be the most likely to represent bona fide interactions. We reasoned that drawing on dozens or hundreds of experiments in any given species would allow us to separate signal from noise to accurately identify these pairs. To minimize unnecessary variation in data pre-processing or quality control, and further increase our ability to separate signal from noise, we re-processed the raw data from all 411

CF-MS experiments through a uniform pipeline, which is a significant strength of this resource. On the other hand, the goal of identifying robust, global interactions is at odds with the application of CF-MS to identify context-specific interactions that might only be detected when focusing on particular biological systems, as we and others have explored[6,9,42–46]. These context-specific interactions will expectantly require more focused analyses of smaller (but biologically and technically homogenous) datasets to detect.

Our meta-analytical approach also meant that we were limited both by the biological scope of the CF-MS datasets that have been deposited to public repositories, as well as the technical limitations of CF-MS itself. For example, whereas CFdb encompasses large resources of uniformly processed CF-MS data from human, *Arabidopsis*, and mouse, there are clearly still opportunities to apply CF-MS more broadly in less-studied organisms (Supplementary Fig. 2c, d). Similarly, while the expansion of CF-MS data in these species allowed us to detect many low-abundance proteins that were not observed in our original resource, this does not negate the more general bias of CF-MS towards highly abundant proteins, which tend to be quantified in the greatest number of fractions (Supplementary Fig. 4b, c).

Beyond the meta-analyses of CF-MS data described here, CFdb also provides a springboard for the development of even more comprehensive and accurate interactome resources by integrating CF-MS data with data from other proteomic techniques. Efforts to this end[47,48] have demonstrated the feasibility of integrating CF-MS data with APMS and proximity labeling datasets to develop integrated interaction networks, using a machine-learning approach similar to that employed here. Future efforts in this vein might additionally draw on thermal protein co-aggregation[49], protein co-abundance[50,51], or structure-based computational inferences[52] to further increase the scope and accuracy of network inference. Notably, combining data from multiple orthogonal techniques could at least partially mitigate some of the inherent biases of CF-MS, such as its preference for stable macromolecular complexes over transient interactions.

With CFdb, we provide a suite of data resources to probe proteome organization across the tree of life. Our results highlight the power of large-scale data integration to answer fundamental questions and create additional proteome-scale resources, and open up new avenues to enhance the analysis of CF-MS datasets by drawing on a wealth of published data.

## Methods

### Comparison of human interactome screens

Data from seven high-throughput screens of the human interactome performed using Y2H or APMS since 2014 was obtained from the supplementary information of the relevant publications or their accompanying websites[53–58]. Protein identifiers were mapped to gene names using flat files downloaded from the UniProt web server, and interacting protein pairs were alphabetized. Overlap between interactome networks was quantified using the Jaccard index.

### MaxQuant searches

We carried out an extensive review of the literature to identify CF-MS experiments that had been published since our original review in May 2020, or which had been overlooked due to differences in terminology. This search identified an additional 205 CF-MS experiments from 41 publications for which raw mass spectrometric data were available as of November, 2022. A handful of experiments performed using data-independent acquisition (DIA) were excluded so as to process all files with an identical pipeline. Raw data were downloaded from the MassIVE or PRIDE repositories, and experimental designs were manually curated to group files into experiments and fractions. Instrument time for each file analyzed was calculated from Thermo RAW files using RawTools[59] (version 2.0.2) and manually retrieved from the corresponding publications for other formats. The complete

list of all CF-MS experiments and raw data files analyzed in this study is provided in Supplementary Data 1.

MaxQuant (version 1.6.5.0) was used to search each experiment against the UniProt complete proteome for the corresponding species, including unreviewed accessions and isoforms, after removal of proteins less than ten amino acids long and supplementation with a list of common contaminants provided by MaxQuant, as previously described[8]. FASTA files are available from GitHub at https://github.com/skinnider/CFdb-searches. Search parameters varied across experiments, but, in general, carbamidomethylation of cysteine was set as a fixed modification, while protein N-terminal acetylation and methionine oxidation were set as variable modifications, and trypsin cleavage was used with up to two missed cleavages. These defaults were then modified as needed for datasets collected with SILAC or TMT labeling, or with proteases other than trypsin. For two in vivo datasets (PXD007288, PXD022309), parameters included semi-specific cleavage with a maximum of two missed cleavages. Instrument settings were adjusted based on the mass spectrometer used to perform the CF-MS experiment. All searches were subsequently repeated with phosphorylation enabled as a variable modification, with default settings including 1% PSM FDR, 1% site-level FDR, a minimum Andromeda score of 40 and a minimum delta score of 6. Code used to download the raw data, create 'mqpar.xml' files and carry out Max-Quant searches is available from GitHub at https://github.com/skinnider/CFdb-searches. Complete MaxQuant outputs are available from PRIDE under the accession PXD042664.

## Preprocessing

MaxQuant outputs ('proteinGroups.txt' files) were preprocessed by removing potential contaminants, reverse hits and proteins identified only by peptides carrying one or more modified amino acids[60]. Protein groups were mapped to gene symbols to enable matching across replicates. For gene symbols that mapped to more than one protein group, only the chromatogram with the fewest missing values was retained. Identifier mapping was performed with flat files downloaded from the UniProt web server. Total numbers of tandem mass spectra and sequenced peptides were obtained from MaxQuant 'summary.txt' files.

## Properties of proteins detected by CF-MS

Human, mouse, and *Arabidopsis* whole-organism protein-abundance estimates in parts per million were obtained from PaxDb (version 4.1)[61]. Estimates of human protein tissue specificity were obtained from the Human Protein Atlas (version 22.0, file 'normal_tissue.tsv.zip') based on the number of tissues in which a given protein was found to be detectably expressed[62]. Coverage of protein complexes was assessed with respect to the core set of protein complexes from CORUM version 3.0 (file 'coreComplexes.txt'), with redundant entries removed[10]. GO terms enrichment analysis of complex proteins detected by CF-MS exclusively in the updated resource was performed using the 'GOstats' R package[63]. Saturation analysis was performed by sampling human, mouse, yeast, and *Arabidopsis* experiments in random order and calculating the total number of proteins quantified in at least one fraction at each step. The process was repeated ten times to estimate variability. A logarithmic curve was fit to the data to project future increases in the human proteome detected by CF-MS with additional experiments. To identify understudied proteins, the number of PubMed IDs associated with each human protein was obtained from PubMed (file 'gene2pubmed.gz').

## Network inference

Interactome networks were inferred for each species represented by at least three CF-MS experiments in CFdb using our previously described computational approach, CFTK[8]. The machine-learning approach is described in detail in the Supplementary Note. Briefly, CFTK infers networks by combining information across multiple CF-MS experiments, using a supervised machine learning paradigm that has been widely applied to map interactome networks from CF-MS data[7,9,27,64–66]. In this paradigm, a machine-learning model is trained to identify interacting protein pairs, using a training set constructed from known protein complexes (for example, those annotated within the CORUM database[10]), such that proteins within the same complex are labeled as interacting pairs and proteins in different complexes are labeled as non-interacting pairs. The data provided as input to the model consists of a series of features computed from one or more CF-MS datasets, each reflecting the similarity between two protein chromatograms in a given dataset. For example, one feature might capture the Pearson correlation between protein chromatograms in a particular CF-MS dataset. Calculating the Pearson correlation across multiple CF-MS datasets would then yield a series of features (one for each experiment), each of which would be provided as input to the machine-learning model. The model is trained to predict whether any given pair of proteins interact, given these features as input. The model is trained in cross-validation to avoid leaking information between the training and test data, and to allow for the possibility that some known complexes may not be assembled in a given dataset. This approach allows CFTK to take as input (i) a set of features computed from one or more CF-MS datasets and (ii) a known set of protein complexes or interacting protein pairs, and return to the user a ranked list of interacting protein pairs, which can then be thresholded at a desired precision.

A critical decision in this workflow entails the choice of features that will be used to represent the similarity of any two protein chromatograms in a given CF-MS dataset. In our original meta-analysis, we established optimal combinations of measures of association and missing value-handling strategies that best separated interacting from non-interacting protein pairs across datasets[8]. Here, we used these optimal combinations to derive features for each CF-MS dataset. The number of optimal features per dataset varied according to the number of CF-MS datasets that had been performed in that species. For species with more than ten datasets, we calculated a single feature per dataset; for species with six to ten datasets, we calculated two features per experiment; and for species with five or fewer datasets, we calculated four features per dataset. The top-four optimal features identified in our previous analysis were, in descending order:

- Distance correlation, with missing values imputed as zeroes
- Weighted cross-correlation, with missing values imputed as zeroes
- Cosine similarity, with missing values imputed as near-zero noise
- Mutual information, with missing values treated as NAs

All features were calculated using CFTK (https://github.com/fosterlab/CFTK), as previously described[8]. Proteins quantified in less than four fractions per dataset were filtered. For eukaryotes, labeled protein pairs were obtained from the CORUM database[10] and ortholog mapping was performed using the eggNOG 'euk' database with the eggnog-mapper tool (version 1.0.3)[67]. For prokaryotes, labeled protein pairs were obtained from the EcoCyc database[34] (file 'protcplxs.col') and ortholog mapping was performed using the eggNOG 'bact' database.

## Network evaluation

Precision was calculated at each point in the ranked interaction list using the labeled protein pairs in that species, and these curves were used to compare the performance of the CFdb interactomes with high-throughput screens in human, mouse, *E. coli*, and yeast. The same "gold standard" set of positive and negative interactions was generated from the CORUM database of curated mammalian protein complexes, and applied to all of the CF-MS and published networks analyzed in this study. Positive examples were defined as pairs of proteins that are part of the same complex ("intra-complex" interactions), whereas negative

examples were defined as pairs of proteins that are both in the set of complexes but not part of the same complex ("inter-complex" interactions). Networks were thresholded at 50% precision, with the understanding that this would provide a highly conservative estimate of true precision: following standard practice within the field, we treat pairs of CORUM proteins found in different complexes as true negatives, meaning that true negatives outnumber true positives by a large margin, and likely include a significant fraction of protein pairs that do participate in interactions that are not captured by the CORUM database[8]. Therefore, to further contextualize the performance of the CFdb interactomes, we drew on multiple orthogonal lines of functional genomic evidence, which were as follows:

- The functional coherence of each network, defined as the degree to which the function of any given protein can be predicted from those of its interacting partners, based on the principle of 'guilt by association'[68]. Briefly, each protein in the network is annotated with its known functions (here, GO terms), and a subset of these labels are then withheld. A neighbor-voting algorithm is then employed to predict functions for the withheld proteins by assigning a score for each GO term that represents the proportion of the protein's interacting partners annotated with the same term. This process is repeated in three-fold cross-validation, and the mean AUC over cross-validation folds is computed for each GO term. A high AUC is characteristic of networks in which proteins that share biological functions tend to be physically connected. Functional coherence analysis was carried out using the 'EGAD' R package[69], filtering GO terms annotated to less than ten or more than 100 proteins. For pairwise comparisons, only GO terms that met this criterion in both networks were included. GO annotations supported only by evidence codes ND, IPI, IEA, and/or NAS were filtered.
- The tendency for interacting proteins to display correlated patterns of abundance across large-scale proteomic datasets. In human, we used the same proteomic datasets as in our previous analysis, namely a meta-analysis of 294 biological conditions using SILAC proteomics in the ProteomeHD resource[50] and a proteomic dataset from cancer cell lines[70]. In mouse, we used two maps of the mouse tissue proteome[71,72]. Coexpression was quantified using the Pearson correlation.
- The tendency for interacting proteins to colocalize to the same subcellular compartments in subcellular proteomic datasets[73,74]. Colocalization was quantified using the Pearson correlation.
- For human networks, we additionally quantified the tendency for genes associated with the same human diseases to physically interact. Disease gene annotations were aggregated from multiple sources, including Online Mendelian Inheritance in Man (OMIM)[75], the NCBI Phenotype-Genotype Integrator (PheGenI)[76], the Mouse Genome Database (MGD)[77], DisGeNET[78], and Menche et al.[79]. Connectivity between disease genes was quantified using the same cross-validation framework as described above for network functional coherence, but here predicting withheld disease genes based on a protein's neighbors in the interactome network.

## Phosphorylation

To identify phosphopeptides in CF-MS data, all 411 CF-MS experiments were subjected to a second MaxQuant search with STY phosphorylation set as a variable modification. MaxQuant outputs ('Phospho(STY) Sites.txt' files) were preprocessed by removing phosphosites from reversed peptides or potential contaminants. The intensity of each phosphosite in each fraction, as well as the ratio between modified and unmodified phosphosite intensities (phosphorylation stoichiometry), were extracted. For datasets collected with SILAC or dimethyl labeling, we additionally extracted the isotopologue ratio of each phosphosite. For TMT datasets, reporter intensities for phosphosites were not output

by MaxQuant, meaning that only aggregate measurements of phosphosite intensity and stoichiometry over all experiments in the TMT plex could be obtained. Phosphosite intensity, stoichiometry, and isotopologue ratio chromatograms for all 411 experiments are available from Zenodo ("Data availability").

To investigate the functional relevance of phosphosites quantified by CF-MS, we obtained phosphosite functional scores for 116,258 scored human sites from Ochoa et al.[17]. Briefly, these scores reflect an estimate of the importance of a given phosphosite for organismal fitness derived from a machine-learning model trained to identify phosphosites with a known regulatory role. The model integrates 59 proteomic, structural, regulatory and evolutionary features, including protein and phosphosite abundance, tissue specificity, evolutionary age, kinase specificity, and presence of neighboring PTMs, among others. Scored phosphosites were detected in a meta-analysis of 6801 phosphoproteomic experiments that specifically enriched for phosphorylated peptides. Functional scores were then compared against the number of fractions or experiments in which a given phosphosite was detected in CFdb. We noted that the median functional score of phosphosite detected in at least 5 fractions was comparable to that of known regulatory sites as shown in Ochoa et al. Based on this observation, we identified phosphosites detected in 5+ fractions in all other species in the dataset to provide a prioritized list of candidate regulatory phosphosites based on CF-MS data integration. We also computed the fraction of tyrosine phosphosites among phosphosites detected in 5+ fractions, and compared this to the fraction in the Ochoa et al. meta-analysis. Moreover, we calculated the enrichment of phosphotyrosine-containing proteins among CORUM complex proteins as compared to proteins phosphorylated on serine or threonine residues, using the Ochoa et al. dataset, and confirmed that this relationship remained significant when adjusting for protein abundance estimates from PaxDb using multivariable logistic regression.

To characterize the structural, evolutionary, and signaling properties of phosphosites detected in at least five CF-MS fractions, we used a dataset of phosphosite properties compiled by Ochoa et al.[17]. Mass spectrometric properties (number of spectral counts, number of biological samples, and maximum localization probabilities) for each phosphosite were based on an independent resource of 6801 phosphoproteomics experiments, in which no samples overlapped with those used in CFdb. Disordered protein regions were identified with DISOPRED[80]. Phosphorylation hotspots[21] were identified as protein domains with significant enrichment in phosphorylation over random, using code available from https://github.com/evocellnet/ptm_hotspots. Phosphosite matches to known protein kinase motifs were assessed using two complementary strategies, first by compiling position weight matrices based on kinase-substrate relationships in the PhosphoSitePlus database[16], and second by using NetPhorest[81] to predict whether acceptor residues matched to a kinase recognition motif or phosphorylation-dependent binding domain. Separately, we performed a kinase-substrate enrichment analysis by using the hypergeometric test to identify kinases enriched among phosphosites identified in five or more CF-MS fractions compared to the background of phosphosites ever identified by CF-MS, using annotated kinase-substrate relationships from PhosphoSitePlus[16]. The ancestral age of human phosphosites was reconstructed by combining phylogenetic information with cross-species phosphoproteomics, using a previously described method[22] (code available from https://github.com/evocellnet/ptmAge).

## Honey bee CF-MS experiments

Honey bees were emerged from a brood frame in a 33°C incubator. Eight honey bee midguts per sample were dissected on ice with forceps and rinsed with cold 1× PBS. The tissues were then disrupted on ice with a Dounce homogenizer in 1 mL of lysis buffer [size exclusion chromatography (SEC) buffer: 50 mM potassium chloride, 50 mM

sodium acetate and 50 mM Tris Base at pH of 7.2, with 1× HALT protease inhibitor (Thermo Fisher)] for three 1 min intervals using a tight pestle. The lysates were transferred into thick-walled ultracentrifuge tubes (Seton) and precleared of insoluble components by ultracentrifugation at 4℃ for 15 min at 100,000 × g. The supernatants were concentrated to ~200 μL using ultrafiltration spin columns with 100 kDa molecular weight cut-off membranes (Vivaspin and Amicon) to isolate protein complexes. 1 g of each sample was then fractionated by size exclusion chromatography on a 1200 series HPLC instrument (Agilent Technologies) using two connected columns (Sepax SRT-C SEC-500, 5 μM 500 Å 7.8 × 300 mm) equilibrated in SEC buffer. The instrument was cooled at 6℃ and 80 fractions were collected from 20-30 min in a 60 min isocratic method with a flow rate of 0.5 mL/min. Fractions were subsequently pooled to 40 fractions prior to protein digestion.

## Protein digestion and cleanup

200 μL of the fractionated samples were constituted with a freshly prepared mixture of urea and thiourea to a final concentration of 6 M and 2 M, respectively. Proteins were then reduced with 2 μg of dithiothreitol and alkylated with 10 μg of iodoacetamide by incubation at room temperature for 30 and 20 min, respectively. Following this, 1 μg of LysC/trypsin (Promega) was added to each sample and incubated for 3 h before diluting the samples with 4 volumes of digestion buffer (50 mM $NH_4HCO_3$). Another 1 μg of LysC/trypsin was then added to digest the proteins overnight. Peptides were acidified in trifluoroacetic acid and desalted using home-made C18 STAGE-TIPS[82] prior to MS analysis.

## LC-MS/MS analysis

Peptides were reconstituted in buffer A (0.5% acetonitrile and 0.1% formic acid in Baker water) for LC-MS/MS analysis. After their concentrations were determined using a NanoDrop spectrophotometer (Thermo Fisher), 100 ng of each peptide sample was injected onto a nanoElute UHPLC system (Bruker Daltonics) and separated using an Aurora Series Gen2 (CSI) 25 cm × 75 μm 1.6 μm FSC C18 column; with Gen2 nanoZero and CSI fittings (Ion Opticks, Australia) at a flow rate of 0.3 μL/min and temperature of 7℃. The column was heated to 50℃ and coupled to a trapped ion mobility-time of flight (timsTOF) Pro mass spectrometer (Bruker Daltonics) which was operated in DIA-PASEF mode. A standard 30 min gradient was run from 2-12% buffer B (0.1% formic acid in 99.4% acetonitrile) over 15 min, then to 33% B from 15 to 30 min, 95% B over 0.5 min and finally held at 95% B for 7.72 min. Prior to each run, the analytical column was conditioned using four column volumes of buffer A.

The timsTOF Pro was set to Parallel Accumulation-Serial Fragmentation (PASEF) scan mode for data independent acquisition (DIA) scanning 100-1700 m/z range. The capillary voltage was set to 1800 V, drying gas to 3 L/min and drying temperature to 180℃. The MS1 scan was followed by 17 PASEF ramps with 22 non-overlapping 35 m/z isolation windows, spanning an m/z range of 219.5-1089.5 (Supplementary Data 1). In the TIMS, ion mobility range (1/k0) was set to 0.70-1.35 V·s/cm$^2$, with 100 ms ramp time and accumulation time (100% duty cycle), and ramp rate of 9.42 Hz to yield 1.91 s of total cycle time. The collision energy was ramped linearly as a function of mobility from 27 eV at 1/k0 = 0.7 V·s/cm$^2$ to 55 eV at 1/k0 = 1.35 V·s/cm$^2$.

## Protein identification and quantitation

Proteins were identified and quantified using Data-Independent Acquisition by Neural Networks (DIANN, version 1.8.1)[83]. A 'library-free' search was done via in silico spectral library construction using the DIA runs and a FASTA file containing the honey bee proteome (UP000005203) as well as the proteomes of common honey bee pathogens, all of which were downloaded from UniProt. Cysteine carbamidomethylation was enabled as a fixed modification with

N-terminal methionine excision enabled and up to one missed cleavage. The output files were filtered to a 1% FDR. Non-honey bee proteins and contaminants were filtered prior to further analysis. The FASTA file, raw mass spectrometry data and all DIA-NN output files are available at PXD042820.

## Data augmentation

Network inference from CF-MS data by supervised machine learning requires the definition of a training set of known protein complexes. While large catalogs of protein complexes are available for model organisms such as human and yeast, such catalogs may be unavailable for non-model organisms. If this is the case, few labeled protein pairs will be available to train the machine-learning model, which may in turn compromise the size or quality of the resulting network.

To date, the prevailing paradigm within the field has been to analyze each dataset in isolation. We hypothesized that injecting small amounts of labeled protein features from published datasets could improve the performance of the classifier in cases where few labeled protein pairs are available by encouraging the classifier to learn relationships between chromatographic similarity and interaction probability that are generalizable across datasets. Importantly, this data augmentation strategy does not involve supplementing the CF-MS data with external evidence that any given pair of proteins interact (i.e., genomic data integration[84]), but instead involves providing the classifier with additional labeled examples that augment the labeled protein pairs within a given set of CF-MS experiments (Supplementary Fig. 10a). These injected examples are then discarded after model training, such that they do not influence the calculation of precision at any point in the ranked list of protein pairs.

To implement this data augmentation strategy, we computed the distance correlation for all labeled protein pairs across 206 human and mouse CF-MS experiments. Protein complexes from CORUM were mapped to their honey bee orthologs using InParanoid[85], and feature calculation and labeling was performed for the eleven honey bee CF-MS experiments as described above. After extracting labeled protein pairs in order to train the model, a random fraction of labeled protein pairs from the external CF-MS experiments were injected into the training data, in proportion with the total number of labeled pairs. Features from external datasets were mapped at random onto individual honey bee experiments. The model was trained in cross-validation on the augmented data, and labeled protein pairs were then discarded prior to computing the precision only on honey bee protein complexes. The random forest model and cross-validation procedure were thus identical to that used throughout the manuscript, and network inference differed only insofar as features from a subset of labeled protein pairs in human or mouse were used to augment the training data from honey bee. Performance was found to be optimized with relatively low proportions of labeled data injection (on the order of 25-50%); the results in the main text show augmentation by a factor of 33%.

To further corroborate the quality of the honey bee interactome mapped with or without data augmentation, we performed several additional analyses. First, the functional coherence of the honey bee interactome was calculated using EGAD as described above, except that GO annotations were not filtered by evidence code due to the sparse annotation of the honey bee genome. Second, the co-expression of interacting protein pairs was calculated across a large honey bee proteomic dataset[26]. Third, interacting protein pairs were mapped to their orthologs in *Drosophila* using InParanoid, and the co-fractionation of these orthologs was calculated in a set of four published CF-MS experiments in fly[27].

## Network inference without a training set of protein complexes

Last, we explored the possibility of using CFdb as a training dataset to enable network inference in species for which few protein complexes

are known or can be inferred through orthology mapping. We focused on four understudied prokaryotes (*Anabaena* sp. 7120, *Cyanothece* sp. ATCC 51142, *Kuenenia stuttgartiensis*, and *Synechocystis* sp. PCC 6803) represented in CFdb for which between 109 and 126 proteins, and just 158 to 190 intra-complex interactions, could be mapped from the EcoCyc protein complex database to one-to-one orthologs in each species. We trained random forest classifiers with 100 trees on labeled protein pairs from 206 human and mouse CF-MS datasets. Labeled pairs from human and mouse were randomly assigned to replicates so as to match the number of replicates acquired in each prokaryote (two for *Cyanothece* sp. ATCC 51142 and *Kuenenia stuttgartiensis*, four for *Synechocystis* sp. PCC 6803, and six for *Anabaena* sp. 7120). Random forest models trained on human and mouse data were then used to score protein pairs in each prokaryote. Because of the limited number of true-positive and true-negative protein pairs available to calculate precision, we instead computed the area under the ROC curve over all scored protein pairs. To assess the functional coherence of the resulting networks, we thresholded the ranked list of scored protein pairs to retain only the top-5000 or top-10,000 interactions[86], then used EGAD to quantify the connectivity of proteins annotated with each GO term, as described above except that GO terms supported only by the IEA evidence code were not filtered.

## Visualization

Throughout the manuscript, box plots show the median (horizontal line), interquartile range (hinges) and smallest and largest values no more than 1.5 times the interquartile range (whiskers).

## Reporting summary

Further information on research design is available in the Nature Portfolio Reporting Summary linked to this article.

## Data availability

A list of all raw mass spectrometry files analyzed in this study and their accession numbers in PRIDE or MassIVE repositories is provided in Supplementary Data 1. Source code used to download and re-analyzed published CF-MS data is available at https://github.com/skinnider/CFdb-searches. Processed chromatograms, phosphosite chromatograms, and MaxQuant 'proteinGroups.txt' and 'Phospho(STY) sites.txt' files are available via Zenodo at https://doi.org/10.5281/zenodo.8008094. Pre-calculated features for each species are available via Zenodo at https://doi.org/10.5281/zenodo.8005773. Predicted interactomes for each species are available via Zenodo at https://doi.org/10.5281/zenodo.10038713. Complete MaxQuant outputs for all 411 CF-MS experiments have been deposited to the PRIDE repository[87] with the dataset identifier PXD042664. Honey bee CF-MS datasets have been deposited to the PRIDE repository with the dataset identifier PXD042820. Other databases used in the study were as follows: PaxDb (https://pax-db.org/), Human Protein Atlas (https://www.proteinatlas.org/), CORUM (http://mips.helmholtz-muenchen.de/corum/), EcoCyc (https://ecocyc.org/), PhosphoSitePlus (https://www.phosphosite.org/homeAction.action), and BioGRID (https://thebiogrid.org/). Source data are provided with this paper.

## Code availability

CFTK is available from GitHub at https://github.com/fosterlab/CFTK. Source code used to download and re-analyzed published CF-MS data is available at https://github.com/skinnider/CFdb-searches. Source code used to carry out analyses presented in the paper, including relevant intermediate data files, is available at https://github.com/skinnider/CFdb-analysis.

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

## Acknowledgements

This work was enabled in part by the support provided by WestGrid and Compute Canada and through computational resources and services provided by Advanced Research Computing at the UBC. M.A.S. acknowledges support from a Vancouver Coastal Health–CIHR–UBC MD/PhD Studentship. We thank J. Moon for advice on MaxQuant searches and J. Rogalski, J. Yuan, and R. Moravcova for assistance with the mass spectrometry.

## Author contributions

M.A.S., M.O.A., and L.J.F. designed experiments. M.A.S. and M.O.A. performed experiments. M.A.S. wrote the manuscript.

## Competing interests

The authors declare no competing interests.
