## [Peer Review File · Nature Communications]

REVIEWER COMMENTS

Reviewer #1 (Remarks to the Author):

Authors present an expanded data resource of CF-MS data sets and CFTK analysis of the merged data. They characterized additional interaction candidates in terms of protein abundance range and post-translational modification (phosphorylation), and demonstrated the use of the resource via the analysis of honey bee interactome prediction using a transfer learning approach. Meta-analysis of CF-MS data is informative as the same authors argued in their previous article. Although I have some objections to pooling CF-MS data with laboratories designing fractionation at different resolutions (and I admit rigorous scoring methods for interaction candidates is the key here), generally this is a growing area and I believe the article deserves a chance to be thoroughly vetted.

Major comments

1. Precision and recall analysis throughout the paper: In benchmarking the interaction candidates from CF-MS experiments against the published AP-MS data sets, one has to carefully define what constitutes the denominators of precision and recall, in formula format if possible, and what were considered as true positives, false negatives, and false positives have to be stated explicitly. This applies to the numerator, too. Methods section did not have this description if my eyes didn't miss them.
2. In the phosphosite analysis section, authors note that their data captures more pY's than other interactome data sets. Another interpretation of a higher proportion of pY could be decreased proportion of pS, certainly naturally disproportionate to the known proportion of phosphosites in databases such as PhosphoSitePlus (<https://www.phosphosite.org/staticSiteStatistics>). But in the end, what does this mean and why is it important for your data resource?
3. On the argument on page 3, right column, prioritizing functional phosphosites. I am not really convinced the functionality of phosphosites is a matter to be resolved with CF-MS data (nor AP-MS). In fact, I would have been more agreeable to call phosphosites more likely to be functional if the sequenced aligned, site-level evidence suggests co-fractionation on orthologs between species. But the way the authors describe, e.g. five fractions or above, in collection of data sets with such different numbers of fractions, doesn't sound convincing. I do not fully support that such a large proportion of a short paper be allocated to this analysis. I am instead alarmed by the level of possible misinterpretation by the readers or users of the data resource as far as phosphorylation is concerned.

4. Honey-bee experiment section can be better described in detail, why was it almost shoved to the end as if it was a supplementary data set?

5. Define transfer learning clearly. What is the exact learning algorithm represented by “Model” in Supp Figure 10? The paper needs at least an independent supplementary information document detailing the method, with exact formula and workflow diagram – I was completely lost. At the moment, the “Transfer learning” section left me with a number of questions at best. Is it CFTK being trained on other data sets and infusing labelled protein pairs into the samples for the improved score calculation on the test (honey bee) samples?

6. Lastly, why was the article written in such a compact format? There is a lot more to describe in my opinion, especially the details of transfer learning. Phosphosite part is honestly not that essential and doesn't add much value to the data resource.

Minor comments

1. Mean Jaccard index of 0.062 and supplementary information. It is convenient for the authors to report that, but how was the index calculated for external data sets with different bait proteins in AP-MS or Bio/turboID experiments? This calculation is also convoluted by different protein identification depths of experiments.

2. How were the ppm's computed in Figure 1d? Just LFQ divided by the sum of all LFQ in each sample?

3. It will be helpful to educate the readers briefly on how the true false positives are controlled, that is, whether you consider all co-fractionated proteins as “physically interacting” in a single data set, and then they are subjected to CFTK for scoring thereafter? Is there a measure of false discovery rates in CFTK?

4. Precision values for individual interactions in the supp tables – how do you calculate “precision” for each protein? Precision is a metric associated with a particular score threshold in classification, rather than a data point specific measure. Can the authors add some annotation of individual columns in all supplementary tables?

Reviewer #2 (Remarks to the Author):

Sikinnider et al. reanalyzed 411 CFMS datasets, including more than 20K fractionations, to generate protein interaction across species. It also digs information on the phosphorylation modification of interacting proteins. It is quite a helpful resource, not only providing protein networks of 31 species but can also help analyses of other non-model organisms.

1. Except for the increase in database size compared to the last version, what's the most significant difference between these two resources? After reconstructing the protein interaction, what's the PPI overlap between these two networks?
2. The author mentioned "a mean Jaccard index of just 0.062 between any pair of the network". What is the overlap between the new network and other datasets, like BioPlex? If the overlap is still low, can it take the APMS and Y2H data into the CFMS data to achieve a more complicated map?
3. It's great that the CF-MS dataset and transfer learning can benefit the construction of honey bee interactome. As we can see, most adding datasets are from humans, and transfer learning also uses human and mouse datasets. How about plants and fungi? Can this dataset help enhance the analysis of those species? Can this dataset also be used to analyze the protein network of prokaryotes, such as some new environmental or gut microbiology?
4. What's the difference between supplementary figure 3d and the pink line in supplementary figure 3d? They look identical. If they are the same, suggest combining these two figures. The same as supplementary figure 3 f and g, i and 3j. Supplementary Fig 3e shows that 87% more proteins in 500+ fractions. However, I didn't see such a big increase from Supplementary Fig 3c. How were those values calculated? It's better to provide the original information in the supplementary tables.
5. Supplementary Table 2 only provide the PPIs of human and mouse. Please add PPIs of all species and mark which PPI is also found in version 1 and which is new.
6. Across all fractionation dataset, are there any PPIs changes because of phosphorylation modification?

Reviewer #3 (Remarks to the Author):

The manuscript “Mapping protein states and interactions across the tree of life” uses established datasets of CF-MS data to increase coverage of protein interactions and provide more reliable consensus datasets for the broad overview of protein relationships across species, expanding from previous work from a little over 200 datasets to over 400, including (and importantly) expanding to new species. Authors note that many of the interactions that are novel to this dataset over the previous are associated with low abundance or are tissue-specific, which historically has limited protein interaction detection. Interactions found in the new network have a greater overlap with published studies than the original network (fig 5), which suggests the network is improving, and co-localization and co-expression data is promising. Overall, the resource looks to be significant and useful to the scientific community.

Issues:

1. The phosphosite functional relevance prediction via machine learning is interesting as a source of hypothesis-generation or prioritization for further research, however I find certain things less compelling. Authors note that detected phosphosites are more likely to have some source of data for machine learning prediction of function, and I am certain that things that are detectable are statistically more frequently discussed or partially characterized in some way than things that are undetectable, but a better evaluation of this might be to examine whether the specific prediction of function via machine learning matches established or expected function from some other screen or database, such as if cell cycle or DNA damage signaling or stress response kinase targets match the machine learning predictions here assigned.

2. The attempt to use homologous interactions and limited CF-MS screens to improve CF-MS data for honeybees was particularly interesting for the application of this system to non-well characterized species, something that will no doubt become more significant to the research community as proteomics work expands to new organisms. However, it would be more convincing if there was biological data presented beyond correlations with expression or localization, such as example complexes or a promising established structure being identified. Moreover, providing some measures to determine the effectiveness of their established pipeline in other non-model organisms would strengthen the findings.

3. The authors used high throughput data from 2014, it would be nice to discuss how the potential biases introduced by the following factors (in the studies) were mitigated or considered during their analysis:

- Quality of data because of variation in experimental protocols between studies.

- Biological systems are dynamic and may change over time and interactions might be context dependent or transient.

- Y2H and AP-MS are two different methods for interaction detection having their own merits and demerits – Y2H might have missed interactions in the context of human cells and AP-MS might have false positive interactions due to purification processes.

4. I wonder why in the original meta-analysis (“version 1”) no phosphosite was detected. All the sites in (Fig 1b last graph) are detected in the version 2 analysis (in pink).

5. The improvement (Fig. 1c-e) after doubling the resource size might still be biased – towards certain species and/or proteins (less abundant or less studied).

6. Findings in this paper such as interactions, and functional relevance of phosphosites (Fig. 2c and d) should be confirmed with established experimental methods (co-immunoprecipitations, functional assays, etc.) to ensure accuracy and reliability.

7. The article lacks an in-depth discussion of potential technical limitations associated with CF-MS and the impact of data preprocessing, quality control, and potential biases on the resulting interaction networks.

8. The article focuses predominantly on CF-MS and its advantages, but it could benefit from discussing how the presented approaches and insights could complement or be integrated with other proteomic techniques.

9. However, comparing and integrating CF-MS data from different studies can be challenging due to variations in experimental protocols, data acquisition methods, and data processing approaches. This should be specified in the text, as well as how they overcome these challenges. Moreover, the human or mouse CF-MS data are most likely retrieved from different cell lines or tissues. Considering the proteome profile differences (i.e., proteins are expressed at different levels depending on the human cell model studied) for each cell line/tissue, a global human interactome could be limiting. Since for each cell-line/tissue the protein levels are different, protein-protein interactions vary as well. This should be addressed, specifying the possible differences in the interactome depending on the tissue analyzed.

Minor questions:

1. I am unsure as to the value of 'instrument time' as a metric of the meta-analysis.

2. The significant increase in tyrosine residues as phosphorylation sites could do with more comment, as this is a departure from expectations (as authors note). Why are so many more tyrosine sites now being identified? This would only require a minor hypothesis.

3. Missing caption of the panels f and g in the supplementary Fig. 4.

Response to reviewers

We thank the reviewers for their careful review of our manuscript, “**Mapping protein states and interactions across the tree of life**,” and for their thoughtful feedback. We were grateful for their enthusiasm for our work, and are glad to be able to respond to all of the points raised in their reviews. We have highlighted all the changes to the text in the accompanying manuscript, but also included the new text in the following responses for convenience. Throughout this document, reviewer comments are shown in **blue**, with our own response in **black**, and changes to manuscript text or figure captions in **red**.

Reviewer #1 (Remarks to the Author):

Authors present an expanded data resource of CF-MS data sets and CFTK analysis of the merged data. They characterized additional interaction candidates in terms of protein abundance range and post-translational modification (phosphorylation), and demonstrated the use of the resource via the analysis of honey bee interactome prediction using a transfer learning approach. Meta-analysis of CF-MS data is informative as the same authors argued in their previous article. Although I have some objections to pooling CF-MS data with laboratories designing fractionation at different resolutions (and I admit rigorous scoring methods for interaction candidates is the key here), generally this is a growing area and I believe the article deserves a chance to be thoroughly vetted.

Major comments

1. Precision and recall analysis throughout the paper: In benchmarking the interaction candidates from CF-MS experiments against the published AP-MS data sets, one has to carefully define what constitutes the denominators of precision and recall, in formula format if possible, and what were considered as true positives, false negatives, and false positives have to be stated explicitly. This applies to the numerator, too. Methods section did not have this description if my eyes didn't miss them.

We apologize that these points were unclear. Based on this comment, as well as the reviewer's other comments below related to transfer learning, precision, and false discovery rates, we have added a supplementary note to the paper (as requested in one of the following comments) that describes each stage of our machine-learning pipeline in detail. This includes the definitions of the positive and negative interactions that were used to calculate precision:

A gold standard reference set of positive and negative interactions was generated from the CORUM database of curated mammalian protein complexes. Positive examples were defined as pairs of proteins that are part of the same complex (“intra-complex” interactions), whereas negative examples were defined as pairs of proteins that are both in the set of complexes but not part of the same complex (“inter-complex” interactions).

These definitions were also provided in the Methods section itself, but perhaps not as clearly as they should have been. We revised the main text Methods to state more prominently (i) the definitions of true and false positives and (ii) that the same definitions were applied throughout the study:

Network evaluation. Precision–recall curves was calculated at each point in the ranked interaction list using the labelled protein pairs in that species, and these curves were used to compare the performance of the CFdb interactomes with high-throughput screens in human, mouse, *E. coli*, and yeast. The same “gold standard” set of positive and negative interactions was generated from the CORUM database of curated mammalian protein complexes, and applied to all of the CF-MS and published networks analyzed in this study. Positive examples were defined as pairs of proteins that are part of the same complex (“intra-complex”

interactions), whereas negative examples were defined as pairs of proteins that are both in the set of complexes but not part of the same complex (“inter-complex” interactions). [...]

With respect to our use of the term ‘recall’ in the legend of **Fig. 1f,g** and **Fig. 2h**, this was an inaccurate use of the term and we thank you for pointing this out. The x-axis does not show recall but rather shows the total number of protein interactions in the network (as labelled in the x-axis of the figure itself). We have corrected this by removing the term ‘recall’ throughout the manuscript. For example, the revised **Fig. 1** legend now reads:

Fig. 1 | A harmonized resource of CF-MS data charts protein abundance and interactions across the tree of life.

[...]

f, Precision of the human interactome inferred by meta-analysis of CF-MS experiments in CFdb as compared to our original meta-analysis, for interaction networks of a given size.

g, Precision of the human interactome inferred by meta-analysis of CF-MS experiments in CFdb for interaction networks of a given size, as compared to six high-throughput screens of the human interactome using Y2H or AP-MS.

2. In the phosphosite analysis section, authors note that their data captures more pY’s than other interactome data sets. Another interpretation of a higher proportion of pY could be decreased proportion of pS, certainly naturally disproportionate to the known proportion of phosphosites in databases such as PhosphoSitePlus (<https://www.phosphosite.org/staticSiteStatistics>). But in the end, what does this mean and why is it important for your data resource?

First, we want to clarify that the background in this analysis (i.e., the ‘observed’ column in the original **Fig. 2b**) is not a database of literature-curated phosphosites such as PhosphoSitePlus, but is instead drawn from the largest existing meta-analysis of phospho-enriched proteomic data, performed by Ochoa et al. (*Nat. Biotechnol.* 2019, doi: 10.1038/s41587-019-0344-3). With respect to the reviewer’s suggestion, we analyzed the distribution of curated phosphosite residues in PhosphoSitePlus and interestingly, found this to be more similar to our own data, with 16.5% pY sites compared to 15.1% detected by CF-MS. We apologize that the details of this analysis weren’t clear from the text and have revised the manuscript accordingly to provide more detail on the analysis that was performed here (shown below). We also updated the figure to show the proportion of pY residues in PhosphoSitePlus:

Fig. 2 | CFdb prioritizes functional phosphosites and enhances analysis of CF-MS data from non-model organisms.

[...]

b, Proportion of phosphoserine (pS), phosphothreonine (pT) and phosphotyrosine (pY) residues in a large-scale meta-analysis of the human phosphoproteome¹⁶, as compared to phosphosites detected by CF-MS, or phosphosites from the curated PhosphoSitePlus database¹⁵.

More to the point, we took the opportunity to clarify our interpretation of this data. To clarify, the hypothesis that we developed from this observation is that tyrosine phosphosites are overrepresented within protein complexes. This hypothesis led us to carry out a follow-up experiment in which we re-analyzed the Ochoa *et al.* dataset, and found that phosphotyrosines were enriched among protein complexes in their dataset as well. We have clarified our hypothesis and the motivation for carrying out this second enrichment analysis in the revised manuscript. The revised paragraph now reads as follows:

Interestingly, whereas radioisotope¹³ and phosphoproteomic¹⁴ data suggest that phosphotyrosines (pY) account for less than 1% of the phosphoproteome, we observed a substantially greater proportion of pY sites in CF-MS data (**Fig. 2b** and **Supplementary Fig. 8a-c**). Tyrosine residues accounted for 15.1% of the phosphosites detected by CF-MS, similar to the proportion in the curated PhosphoSitePlus database¹⁵ (16.5%) but greater than that observed in the largest meta-analysis of phospho-enrichment proteomic datasets to date (5.1%)¹⁶. Given that most CF-MS protocols enrich for protein complexes by design, this observation raised the possibility that tyrosine phosphosites might be overrepresented within protein complexes. In agreement with this possibility, pY-containing proteins were also enriched among protein complexes in a large-scale meta-analysis of phosphoproteomic datasets¹⁶ (**Supplementary Fig. 8d**), and this enrichment remained statistically significant when controlling for protein abundance ($p = 9.4 \times 10^{-4}$, logistic regression).

3. On the argument on page 3, right column, prioritizing functional phosphosites. I am not really convinced the functionality of phosphosites is a matter to be resolved with CF-MS data (nor AP-MS). In fact, I would have been more agreeable to call phosphosites more likely to be functional if the sequenced aligned, site-level evidence suggests co-fractionation on orthologs between species. But the way the authors describe, e.g. five fractions or above, in collection of data sets with such different numbers of fractions, doesn't sound convincing. I do not fully support that such a large proportion of a short paper be allocated to this analysis. I am instead alarmed by the level of possible misinterpretation by the readers or users of the data resource as far as phosphorylation is concerned.

Thank you for raising these issues. We would certainly hate for readers to misinterpret the phosphorylation data. At the same time, we do feel that this data represents a substantial addition to the resource that will be of value to the community. We believe that we can address this point by being more careful in how we present this data, as we discuss below.

First, we want to underscore why we think our meta-analysis of phosphorylation across >21,000 CF-MS fractions is a unique dataset and a valuable contribution. The resource presented here is one of just a handful of meta-analyses of phosphoproteomic data that have been carried out to date (including also Ochoa *et al.*, *Mol. Syst. Biol.* 2016; Hernandez-Armenta *et al.*, *Bioinformatics* 2017; Ochoa *et al.*, *Nat. Biotechnol.* 2020). Here, we are providing a new, large-scale dataset of protein phosphorylation can be used to answer a number of different biological questions or contribute to the development of new computational tools, such as methods to computationally predict phosphorylation sites or phosphopeptide mass spectra, both of which have been topics of increasing interest in the last few years. This, we feel, is a useful contribution. Moreover, to our knowledge, there is no existing dataset in which phosphorylation has been (meta-)analyzed across a large resource of interaction proteomics data, so our dataset also provides a novel contribution in this respect. We expect that this dataset can provide a springboard to answer numerous questions that are difficult to address systematically with existing datasets—for example, the possibility the reviewer raises of comparing orthologous phosphosites across species.

Second, to corroborate the notion that our criterion of detection in at least five CF-MS fractions is detecting a biologically relevant signal, we carried out a more detailed characterization of the human phosphosites detected in 5+ fractions. This analysis allowed us to establish that these frequently detected phosphosites have a number of unusual features relative to the phosphoproteome average. These include technical characteristics associated with reliable identification, but also a number of interesting biological properties. For instance, phosphosites detected in 5+ CF-MS fractions are more likely to be located in intrinsically disordered regions or phosphorylation ‘hotspots’; more likely to match known protein kinase motifs; enriched for known substrates of several kinases by kinase-substrate enrichment analysis (KSEA); and more likely to be evolutionarily ancient (e.g., conserved across all tetrapods or bilaterians). We present these results in a new supplementary figure, reproduced below:

Supplementary Fig. 9 | Properties of frequently detected phosphosites.

a, Number of MS/MS spectra in which phosphosites were detected in an independent resource of 6,801 phosphoproteomics experiments, in which no samples overlapped with those used in CFdb¹⁶, shown separately for phosphosites detected in 1-4 versus 5 or more CF-MS fractions ($p < 10^{-15}$, Wilcoxon rank-sum test).

b, As in **a**, but showing the number of biological samples in which phosphosites were detected ($p < 10^{-15}$, Wilcoxon rank-sum test).

c, As in **a**, but showing the maximum localization probability with which phosphosites were detected ($p < 10^{-15}$, Wilcoxon rank-sum test).

d, Proportion of phosphosites located within predicted intrinsically disordered regions, shown separately for phosphosites detected in 1-4 versus 5 or more CF-MS fractions ($p = 2.1 \times 10^{-15}$, χ^2 test).

e, As in **d**, but showing phosphosites located within phosphorylation hotspots ($p = 2.0 \times 10^{-8}$, χ^2 test).

f, As in **a**, but showing the maximum kinase position weight motif matrix similarity score for each phosphosite ($p < 10^{-15}$, Wilcoxon rank-sum test).

g, As in **a**, but showing the maximum NetPhorest posterior probability for each phosphosite ($p < 10^{-15}$, Wilcoxon rank-sum test).

h, Kinase-substrate enrichment analysis of phosphosites. Left, number of expected (blue) versus observed (red) substrates of kinases from the PhosphoSitePlus database among phosphosites detected in 5 or more CF-MS fractions, compared to phosphosites ever detected by CF-MS; right, statistical significance of the observed enrichment.

i, As in **d**, but showing the inferred ancestral age of each phosphorylation site ($p = 1.3 \times 10^{-15}$, χ^2 test).

j, Bars, total number of CF-MS fractions collected per major species or taxonomic groups. Text, proportion of fractions per species corresponding to detection in five or more CF-MS fractions.

We also present this data in the revised Results section as follows:

Phosphosites detected in five or more CF-MS fractions had a number of unusual features relative to the phosphoproteome average. In an independent phosphoproteomics dataset, these phosphosites tended to be identified in more samples, by more MS/MS spectra, and with higher localization probabilities, all features characteristic of more reliable identifications (**Supplementary Fig. 9a-c**). They were also more likely to be located in intrinsically disordered regions¹⁹ and in recurrently phosphorylated structural regions (phosphorylation ‘hotspots’²⁰), both of which are known to be enriched for functional phosphosites (**Supplementary Fig. 9d-e**). Frequently quantified phosphosites were more likely to match known protein kinase motifs, and kinase-substrate enrichment analysis identified several kinases whose known substrates were enriched among these phosphosites, with the most significant enrichments for CK2A1, CDK1, and CDK2 (**Supplementary Fig. 9f-h**). Last, estimates of phosphosite evolutionary age²¹ suggested that frequently quantified phosphosites were evolutionarily ancient, with human-specific phosphosites depleted and phosphosites conserved across tetrapods or bilaterians enriched among this set (**Supplementary Fig. 9i**).

Third, we consider the reviewer’s point about using a simple cutoff of five or more fractions in a resource with very different numbers of fractions per species. We agree that this an arbitrary cutoff, and that more generally, CF-MS itself does not provide direct evidence of function. Rather, what our data demonstrates is that phosphosites that are recurrently detected by CF-MS are enriched for functional phosphosites. With respect to the specific cutoff of five fractions, we also wanted to clarify that because other species are represented by fewer CF-MS fractions, this actually reflects a *more* stringent criteria in non-human species: for example, detection in 5+ fractions represents ~0.05% of the human dataset, but ~0.20% of the mouse dataset and ~0.43% of the *C. elegans* dataset. We added a panel to **Supplementary Fig. 9** to more directly visualize this (shown above). Moreover, we revised the relevant section of the Results accordingly to clarify both of these points:

Calculation of phosphosite functional scores requires the laborious integration of proteomic, structural, regulatory and evolutionary datasets, which has so far limited this approach to the human proteome. The observation that phosphosites quantified in at least five CF-MS fractions display functional scores comparable to known regulatory phosphosites suggested that detection by CF-MS could be used to prioritize functional phosphosites across species. To evaluate this possibility, we searched for phosphosites detected in at least five CF-MS fractions in other species, recognizing that imposed a somewhat more stringent criterion because fewer fractions were profiled by CF-MS in species other than human (**Supplementary Fig. 9j**). Beyond human, an average of 543 phosphosites per species met this criterion, yielding a total of 16,832 prioritized phosphosites across 31 species (**Fig. 2e** and **Supplementary Table 4**). Together, these results suggest that CFdb can contribute to prioritization of functional phosphosites across species, although it is important to emphasize that CF-MS does in and of itself not provide direct evidence of function.

Fourth, we made several other changes to the text to address the limitations discussed above in the presentation of our data. We removed the claim about ‘data-driven identification of functional phosphosites’ from the abstract as a potential oversimplification so that we could provide a more balanced presentation in the text:

[...] Meta-analysis of this resource charts protein abundance, phosphorylation, and interactions throughout the tree of life, including a new reference map of the human interactome ~~and a framework for data-driven prioritization of functional phosphosites across 32 species.~~ [...]

We also added a paragraph to the Discussion section articulating what we feel are the strengths and limitations of the resource:

Beyond protein-protein interactions, we also carried out a meta-analysis of protein phosphorylation across all 21,703 fractions. This resource adds to the relatively small number of phosphoproteomic meta-analyses that have been carried out to date^{16,26,27}, and provides a new dataset that can be used to answer a number of different biological questions or contribute to the development of new computational tools, such as methods to computationally predict phosphorylation sites^{28,29} or phosphopeptide mass spectra^{30–32}. This resource is also unique in that protein phosphorylation has rarely been investigated with CF-MS, except by studies that employed bespoke experimental designs¹². Here, we show that, despite the lack of explicit phosphopeptide enrichment in published CF-MS datasets, many phosphosites are detected across dozens or hundreds of fractions. These frequently detected phosphosites have a number of unusual features relative to the phosphoproteome average. They are evolutionarily ancient, enriched in intrinsically disordered regions and phosphorylation hotspots, more likely to match known protein kinase motifs, and enriched for the targets of several protein kinases. These properties are reflected in the assignment of disproportionately high functional scores to frequently-detected phosphosites by one machine-learning approach¹⁶, relative to both phosphosites ever detected by CF-MS and to the human phosphoproteome in general. Together, these observations suggest that CFdb can contribute a new source of data to efforts to prioritize functional phosphosites across species. However, two major limitations of our analysis are important to note: CF-MS does not in and of itself provide direct evidence of phosphosite function, and here we used a simple heuristic of detection in at least five fractions to prioritize phosphosites based on CF-MS. Although this heuristic allowed us to study a number of interesting features of frequently detected phosphosites, integrating data from CFdb with other proteomic, structural, regulatory and evolutionary datasets will provide a more robust basis for phosphosite prioritization in the future.

Collectively, we think these changes provide (i) a more nuanced presentation of the potential utility of our phosphorylation resource, as well as its limitations and (ii) a much more extensive characterization of the phosphosites detected in five or more CF-MS fractions.

4. Honey-bee experiment section can be better described in detail, why was it almost shoved to the end as if it was a supplementary data set?

We originally submitted this manuscript as a Brief Communication to Nature Methods, a format which imposes a strict word limit, and subsequently transferred it to Nature Communications, a process that required us to keep the same format as the original submission. We have taken the opportunity to revise the manuscript to expand on a number of the details discussed above, as well as the points raised by the other two reviewers. This includes a more extensive description of the dataset itself, as well as more detailed characterization of some individual protein complexes and interactions within the honey bee data. The complete revised honey bee section is reproduced below:

To date, the dominant paradigm in the field has been to analyze each newly collected CF-MS dataset in isolation. We asked whether CFdb could augment analyses of smaller-scale CF-MS projects, such as those

carried out within individual laboratories. We reasoned that such approaches could be particularly useful for studies of non-model organisms, in which few protein complexes may be known.

As a proof of concept, we carried out a new set of CF-MS experiments in honey bee (*Apis mellifera*). Honey bees are vital pollinators that play central roles in global agriculture^{24,25}. However, the honey bee interactome remains largely unmapped. This gap reflects a number of challenges that are common to the study of non-model organisms, including a lack of established cell culture systems, limited amenability to genetic manipulation, and incomplete proteome annotation. CF-MS is well-suited to overcome these limitations by enabling interactome mapping under physiological conditions within *in vivo* tissues, and without requiring validated antibodies or the introduction of protein tags.

As a first step towards mapping the honey bee interactome, we profiled the honey bee midgut by CF-MS. The midgut was selected as the primary site of infection for a prevalent honey bee pathogen, *Vairimorpha (Nosema) cerana*, that has been implicated in the collapse of honey bee colonies. Ten CF-MS experiments were performed in which 40 fractions were collected and mass spectrometry data was acquired with data-independent acquisition (DIA), while an eleventh experiment was analyzed using data-dependent acquisition (DDA). An average of 2,092 proteins were quantified per replicate, yielding a total of 319,105 protein quantifications across all 440 fractions.

We then sought to infer protein interaction networks for the honey bee midgut from this dataset. However, of the 5,163 human proteins within the CORUM database¹⁰, only 1,200 could be mapped to a bee ortholog, limiting the amount of training data available for network inference. We therefore devised a **data augmentation** strategy that leveraged CFdb to augment our bee data with labelled protein pairs that were sampled randomly from 206 human or mouse CF-MS experiments (**Supplementary Fig. 11a**). This strategy increased the size of the honey bee CF-MS interactome by 64.7%, while simultaneously improving its functional coherence (**Fig. 2h**, **Supplementary Fig. 11b-c** and **Supplementary Table 5**). Moreover, interacting proteins in either network showed comparable patterns of coexpression across a large honey bee proteomic dataset²⁶, and their fly orthologs displayed comparable patterns of co-elution in an independent CF-MS dataset²⁷ (**Supplementary Fig. 11d-e**). Data augmentation enabled improved coverage of several CORUM protein complexes with one-to-one orthologs in honey bee, such as the 26S proteasome, the LSm2-8 complex, and the 20S methylosome (**Supplementary Fig. 11f**). Beyond better coverage of known complexes, data augmentation also enabled the identification of novel interactions. For instance, we identified an interaction between calreticulin and inositol-3-phosphate synthase. This interaction was not annotated in any species in the BioGRID database, but these proteins were previously found to co-purify in a large AP-MS study of the fly interactome²⁸, supporting the existence of an orthologous interaction in honey bee (**Supplementary Fig. 11g-h**).

5. Define transfer learning clearly. What is the exact learning algorithm represented by “Model” in Supp Figure 10? The paper needs at least an independent supplementary information document detailing the method, with exact formula and workflow diagram – I was completely lost. At the moment, the “Transfer learning” section left me with a number of questions at best. Is it CFTK being trained on other data sets and infusing labelled protein pairs into the samples for the improved score calculation on the test (honey bee) samples?

We apologize that the details of our machine-learning approach were unclear. To remedy this, we have added a supplementary note in which we walk through the network inference pipeline in detail, as discussed above in response to one of the preceding points.

With respect to our model, the general approach presented here is not a new contribution but instead recapitulates the workflow used in our previous meta-analysis. We also note that this is, in turn, an optimized version of the basic machine-learning approach that has been used to infer protein interaction networks from CF-MS since 2012 (see references in the supplementary note). However, we now provide complete details of our model and approach in the supplementary note.

With respect to the transfer learning itself, we can assure the reviewer that we are not simply infusing labelled data into the dataset in order to improve the calculated precision scores. Our idea was simply to concatenate a subset of the features from labelled protein pairs in human or mouse experiments to the feature matrix from the honey bee CF-MS dataset. Our expectation was that augmenting the bee dataset with a certain fraction of labelled protein pairs from human and mouse datasets would encourage the classifier to learn more general features of protein complex co-elution and mitigate overfitting to the relatively small number of labelled protein pairs in the bee dataset. Importantly, these human and mouse intra-complex pairs were removed prior to calculating the precision, so they did not influence the calculation of the precision score. In other words, the random forest model and the cross-validation procedure were identical to those used throughout the manuscript, and network inference differed only insofar as features from a subset of labelled protein pairs in human or mouse were used to augment the training data from honey bee. We revised the Methods section to clarify this point:

To date, the prevailing paradigm within the field has been to analyze each dataset in isolation. We hypothesized that injecting small amounts of labelled protein features from published datasets could improve the performance of the classifier in cases where few labelled protein pairs are available by encouraging the classifier to learn relationships between chromatographic similarity and interaction probability that are generalizable across datasets. Importantly, this **data augmentation** strategy does not involve **augmenting** the CF-MS data with external evidence that any given pair of proteins interact (i.e., genomic data integration⁸⁴), but instead involves providing the classifier with additional labelled examples that augment the labelled protein pairs within a given set of CF-MS experiments (**Supplementary Fig. 10a**). These injected examples are then discarded after model training, **such that they do not influence the calculation of precision at any point in the ranked list of protein pairs.**

We think that some of the confusion might have stemmed from our use of the term ‘transfer learning.’ In reflecting on how to best address this comment, we realized it might be more accurate to refer to this as a *data augmentation* strategy, since the model is not actually being fine-tuned per se. With this in mind, we have revised the manuscript to replace the use of the phrase ‘transfer learning’ with ‘data augmentation’ throughout.

6. Lastly, why was the article written in such a compact format? There is a lot more to describe in my opinion, especially the details of transfer learning. Phosphosite part is honestly not that essential and doesn't add much value to the data resource.

As discussed in response to point #4, the manuscript was initially submitted as a Brief Communication to Nature Methods, which imposes a limit of 1,600 words. Since we no longer find ourselves constrained by this word limit, we have taken the opportunity to revise the manuscript to expand on a number of the details discussed above, as well as the points raised by the other two reviewers. The revised manuscript is also presented in a more conventional (i.e. Introduction, Results, Discussion) format. Of note, this has allowed us to significantly expand our discussion of the limitations of the current study. We hope that these revisions clarify the methodological details of our analysis and what we see as the major contributions (and limitations) of this resource.

Minor comments

1. Mean Jaccard index of 0.062 and supplementary information. It is convenient for the authors to report that, but how was the index calculated for external data sets with different bait proteins in AP-MS or Bio/turboID experiments? This calculation is also convoluted by different protein identification depths of experiments.

We agree and certainty did not mean to imply that the majority of non-overlapping interactions are false positives. The considerations raised by the reviewer (different depths of protein identification, different bait proteins, etc.) also could have affected the overlap between high-throughput screens, and some of these have already been discussed in the literature as potential reasons for the limited overlap. We revised the paragraph in question to raise these and other considerations as shown below:

Cellular processes arise from the dynamic organization of proteins in networks of physical interactions. Significant resources have been devoted to mapping the protein interaction networks of humans and model organisms¹. These networks are widely used for tasks such as protein function prediction, disease gene prioritization, or interpretation of transcriptomic and proteomic datasets^{2,3}. However, questions about the reproducibility of these networks have persisted. Limited overlap between screens performed in different laboratories was noted soon after the first maps of the yeast interactome emerged^{4,5}. Two decades later, large-scale efforts have produced systematic maps of the human interactome that display relatively little overlap with one another, with a mean Jaccard index of just 0.062 between any pair of networks (**Supplementary Fig. 1**). This lack of overlap has been variously attributed to differences in the types of interactions detected by each assay and the proteins targeted by individual screens, variation in experimental protocols or the depth of protein identification, the presence of context-specific interactions, or false-positives and false-negatives in the resulting interactome maps.

Moreover, in response to a comment by reviewer 2, we now show the overlap between the consensus CF-MS interactome and each of these high-throughput screens, in **Supplementary Fig. 5k**. This analysis provides further context for the Jaccard index between pairs of past screens.

2. How were the ppm's computed in Figure 1d? Just LFQ divided by the sum of all LFQ in each sample?

These are ppm estimates from the PaxDb database. PaxDb aggregates protein abundance data from many different techniques and sources, converts these data to parts-per-million estimates, and averages estimates across all datasets per species. Slightly different approaches are implemented based on the origin of the protein abundance data (e.g. label-free quantification vs. spectral counting vs. biochemical techniques); complete details are provided in Wang et al., *Mol. Cell. Proteomics* 2012, doi: 10.1074/mcp.O111.014704. This was previously discussed in the Methods, but we have also revised the figure legend to now also state the source of the protein abundance data:

Fig. 1 | A harmonized resource of CF-MS data charts protein abundance and interactions across the tree of life.

[...]

d, Abundance of human proteins detected by CF-MS in the original meta-analysis or the updated resource, versus those never detected by CF-MS, based on consensus protein abundance estimates from the PaxDb database⁵⁷.

3. It will be helpful to educate the readers briefly on how the true false positives are controlled, that is, whether you consider all co-fractionated proteins as “physically interacting” in a single data set, and then they are subjected to CFTK for scoring thereafter? Is there a measure of false discovery rates in CFTK?

Again, we apologize that the details of our machine-learning approach were not clear. As discussed above, we elaborate on each of these steps in the supplementary note, which we hope is clarifying. Briefly, for each CF-MS experiments, a series of all-by-all pairwise scores were calculated between the elution profiles of all quantified proteins. These calculated scores were then merged to produce a single feature matrix, which was provided to a random forest classifier as input, alongside a set of known protein complexes from CORUM. A pair of proteins was labeled “positive” if both proteins were in the same

CORUM complex, and “negative” if proteins were both in the set of CORUM complexes, but not part of the same complex. The classifier was then trained to distinguish positive and negative examples (that is, interacting versus non-interacting protein pairs). Classifier training was performed in ten-fold cross-validation, both to minimize overfitting and to allow the classifier to make predictions for protein pairs within the training set of known protein complexes. Protein pairs were ranked in descending order by their mean classifier score across all ten folds. Finally, the classifier score calculated for each protein pair in the ranked list was then converted to a measure of precision for each interaction by calculating the ratio of true positives to true positives plus true negatives among interactions assigned that score or greater by the classifier.

With respect to false discovery rates: calculating the precision at each point in the ranked list implicitly allows us to also compute the false discovery rate as the complement of the precision. However, it is important to recognize that the field currently lacks a comprehensive list of ‘true negative’ interactions, in humans or any other species. Like others in the field, we treat pairs of CORUM proteins found in different complexes as true negatives, but some number of these ostensibly true-negative pairs are likely to truly interact, in pairwise interactions or protein complexes that are as-of-yet undiscovered or simply missing from the CORUM database. This gap makes it challenging to estimate the absolute error rate of protein interaction networks, and indeed, estimating the absolute error rate of protein-protein interaction networks is a problem that has attracted extensive discussion. In view of these caveats, we think the most important message to take away from our analysis is the relative performance of the CF-MS interactomes, compared to previous large-scale screens, when evaluated on identical sets of true-positive and true-negative interactions.

4. Precision values for individual interactions in the supp tables – how do you calculate “precision” for each protein? Precision is a metric associated with a particular score threshold in classification, rather than a data point specific measure. Can the authors add some annotation of individual columns in all supplementary tables?

We have clarified this in the supplementary note that accompanies the revised manuscript. In brief, after ranking protein pairs (not proteins) in descending order by classifier score, we compute the precision at each point in the resulting ranked list. This is done by calculating the number of true and false positives among the n top-ranked protein pairs, so that for example, the precision at protein pair 600 is equal to $TP_{1-600} / (TP_{1-600} + FP_{1-600})$, where TP_{1-600} and FP_{1-600} are the number of true positives and false positives among the first 600 top-ranked protein pairs, respectively.

In addition, we have annotated the columns in all of the supplementary tables as requested.

Reviewer #2 (Remarks to the Author):

Sikinnider et al. reanalyzed 411 CFMS datasets, including more than 20K fractionations, to generate protein interaction across species. It also digs information on the phosphorylation modification of interacting proteins. It is quite a helpful resource, not only providing protein networks of 31 species but can also help analyses of other non-model organisms.

Thank you for your positive comments on our manuscript and your thoughtful feedback.

1. Except for the increase in database size compared to the last version, what's the most significant difference between these two resources? After reconstructing the protein interaction, what's the PPI overlap between these two networks?

Thank you for the opportunity to elaborate on the key advances embodied in this resource. To briefly summarize, there are two broad reasons we are excited about this work and feel it represents a major advance over the previous version:

1. The first is that the expanded resource of CF-MS data in CFdb has allowed us to upgrade all of the resources that we presented in our original analysis. In many cases, the implications of these upgrades are dramatic. For example, through meta-analysis of 166 CF-MS experiments, we have produced a map of the human interactome that multiple lines of evidence suggest is now among the highest-quality human interactomes in existence. Similarly, we present the highest-quality map of the mouse interactome, as well as the first CF-MS interactomes for major model organisms not covered in our original resource such as yeast or *E. coli*. These networks include interactions for hundreds of low-abundance, tissue-specific, and/or 'understudied' proteins not captured in our original resource.
2. The second is that the expanded resource has allowed us to take our analysis in several completely new directions. One major new direction involved searching all 411 CF-MS experiments for phosphopeptides, allowing us to compile a resource of ~750,000 phosphosite quantifications across >21,000 interaction proteomics experiments—we are not aware of any resource like this in existence. We show that phosphosites detected by CF-MS have several unusual and biologically interesting features relative to the proteome average. Separately, we also show how CFdb can enable new approaches to the computational analysis of CF-MS data. We are particularly excited about our demonstration that published CF-MS data can be leveraged at a repository scale to help interpret experiments done in individual labs.

We revised and expanded the Discussion section of the manuscript to clarify the major contributions of our manuscript, as well as its limitations, as shown below:

Discussion

The increasing uptake of CF-MS has led to the deposition of hundreds of datasets in public proteomic repositories. However, these datasets are generally collected and analyzed in isolation by individual laboratories. Here, we explored the possibility of aggregating biologically and technically heterogeneous CF-MS data at the repository scale. We re-analyzed 21,703 fractions from 411 CF-MS experiments using a uniform computational pipeline that standardized protein identification, quantification, and quality control. This expanded resource incorporated data from eight new species and substantially expanded the proteome coverage of humans and model organisms by CF-MS. It also dramatically improved our ability to infer protein interaction networks through meta-analysis of all available CF-MS data for any given species. For example, through meta-analysis of 166 CF-MS experiments, we have produced a map of the human interactome that multiple lines of evidence suggest is among the highest-quality interactome maps currently in existence.

Similarly, we present a high-quality map of the mouse interactome derived from meta-analysis of 40 CF-MS experiments, as well as CF-MS interactomes for major model organisms not covered in our original resource such as yeast or *E. coli*. These networks include interactions for hundreds of low-abundance, tissue-specific, and/or understudied proteins not captured in our original resource. We carried out extensive comparisons to other functional genomics datasets that substantiate the quality of the inferred networks at a systems level. However, we caution users of this resource that any particular interaction should in isolation be regarded as putative until confirmed experimentally using an orthogonal technique.

Beyond protein-protein interactions, we also carried out a meta-analysis of protein phosphorylation across all 21,703 fractions. This resource adds to the relatively small number of phosphoproteomic meta-analyses that have been carried out to date^{16,26,27}, and provides a new dataset that can be used to answer a number of different biological questions or contribute to the development of new computational tools, such as methods to computationally predict phosphorylation sites^{28,29} or phosphopeptide mass spectra³⁰⁻³². This resource is also unique in that protein phosphorylation has rarely been investigated with CF-MS, except by studies that employed bespoke experimental designs¹². Here, we show that, despite the lack of explicit phosphopeptide enrichment in published CF-MS datasets, many phosphosites are detected across dozens or hundreds of fractions. These frequently detected phosphosites have a number of unusual features relative to the phosphoproteome average. They are evolutionarily ancient, enriched in intrinsically disordered regions and phosphorylation hotspots, more likely to match known protein kinase motifs, and enriched for the targets of several protein kinases. These properties are reflected in the assignment of disproportionately high functional scores to frequently-detected phosphosites by one machine-learning approach¹⁶, relative to both phosphosites ever detected by CF-MS and to the human phosphoproteome in general. Together, these observations suggest that CFdb can contribute a new source of data to efforts to prioritize functional phosphosites across species. However, two major limitations of our analysis are important to note: CF-MS does not in and of itself provide direct evidence of phosphosite function, and here we used a simple heuristic of detection in at least five fractions to prioritize phosphosites based on CF-MS. Although this heuristic allowed us to study a number of interesting features of frequently detected phosphosites, integrating data from CFdb with other proteomic, structural, regulatory and evolutionary datasets could provide a more robust basis for phosphosite prioritization in the future.

Regarding the second half of your question, the PPI overlap between the human networks is shown in **Supplementary Fig. 5c**. We also characterize the interactions detected exclusively in versions 1 or 2 of the network in the same supplementary figure.

2. The author mentioned “a mean Jaccard index of just 0.062 between any pair of the network”. What is the overlap between the new network and other datasets, like BioPlex? If the overlap is still low, can it take the APMS and Y2H data into the CFMS data to achieve a more complicated map?

This is a good point. To address it, we computed the Jaccard index between the CF-MS interactome and all of the high-throughput screens shown in **Supplementary Fig. 1**. The results are shown below and have been incorporated as the newly added panel **Supplementary Fig. 5k** in the revised manuscript:

Supplementary Fig. 5 | Interactome networks inferred by large-scale meta-analysis of CF-MS data.
[...]

k, Overlap between high-throughput screens of the human interactome²⁹⁻³⁴ performed using Y2H or AP-MS since 2014, as quantified by the Jaccard index and as shown in **Supplementary Fig. 1** but here including the human interactome derived from meta-analysis of 166 CF-MS experiments.

We found the overlap between the CF-MS interactome and previous screens to be relatively low, comparable to the overlap between pairs of previous high-throughput screens (at least, the subset of those did not rely in part on overlapping datasets, e.g. BioPlex 2 versus BioPlex 3). We comment on this in the revised Results section as follows:

[...] Interestingly, interactions found only in the updated network were significantly more likely to overlap with at least one small-scale or high-throughput experiment, supporting the notion that meta-analysis can improve the reproducibility of biological networks ($p = 1.9 \times 10^{-15}$, χ^2 test; **Supplementary Fig. 5i-j**), **although the overlap between the CF-MS interactome and previous high-throughput screens remained relatively low (Supplementary Fig. 5k).** [...]

Separately, regarding the possibility of integrating AP-MS, Y2H, and other interaction proteomics datasets with the CF-MS data described here in order to derive even more accurate and comprehensive interaction networks, we agree that this is an exciting possibility. This kind of integration is perhaps so far best exemplified by the hu.MAP resources (Drew et al., *Mol. Syst. Biol.* 2017 and 2021), which initially integrated data from one CF-MS study (Wan et al., *Nature* 2015) and two AP-MS studies (Huttlin et al., *Cell* 2015 and Hein et al., *Cell* 2015) using a machine-learning pipeline very similar to that used in our study. The authors subsequently developed an updated version of this resource in which they integrated additional AP-MS datasets, two proximity labeling datasets, and an RNA hairpin pulldown dataset to more accurately identify interacting protein pairs. These efforts therefore benefited from integrating a much broader range of proteomic datasets than we have considered here, but they made use of data from relatively few (albeit unusually large-scale) studies.

We expect that the availability of uniformly processed data from hundreds of CF-MS experiments in CFdb provides a springboard for the development of even more comprehensive and accurate proteome resources. These might draw on not just AP-MS and CF-MS but also proximity labelling, thermal proximity co-aggregation (e.g. Jarzab et al., *Nat. Methods* 2020), protein co-abundance (e.g. Kustatscher et al., *Nat. Biotechnol.* 2019), or structure-based computational inferences (e.g. O'Reilly et al., *Mol. Syst. Biol.* 2023) to identify protein interactions. We have revised the Discussion section to comment on these kinds of opportunities. The newly-added paragraph is reproduced below:

Beyond the meta-analyses of CF-MS data described here, CFdb also provides a springboard for the development of even more comprehensive and accurate interactome resources by integrating CF-MS data with data from other proteomic techniques. Efforts to this end^{22,23} have demonstrated the feasibility of integrating CF-MS data with AP-MS and proximity labelling datasets to develop integrated interaction networks, using a machine-learning approach similar to that employed here. Future efforts in this vein might additionally draw on thermal protein co-aggregation²⁴, protein co-abundance^{25,26}, or structure-based computational inferences^{27,28} to further increase the scope and accuracy of network inference. Notably, combining data from multiple orthogonal techniques could at least partially mitigate some of the inherent biases of CF-MS, such as its preference for stable macromolecular complexes over transient interactions.

3. It's great that the CF-MS dataset and transfer learning can benefit the construction of honey bee interactome. As we can see, most adding datasets are from humans, and transfer learning also uses human and mouse datasets. How about plants and fungi? Can this dataset help enhance the analysis of those species? Can this dataset also be used to analyze the protein network of prokaryotes, such as some new environmental or gut microbiology?

Thank you for this suggestion. We took particular note of the possibility that CFdb could contribute to the analysis of protein interaction networks for understudied prokaryotes. Indeed, there are four such species (*Anabaena* sp. 7120, *Cyanothece* sp. ATCC 51142, *Kuenenia stuttgartiensis*, and *Synechocystis* sp. PCC 6803) in CFdb, for which few if any protein complexes are known. Moreover, attempts to map protein complexes from *E. coli* to their one-to-one orthologs in these species yield extremely small training datasets with just 158 to 190 true positive interactions, which makes standard supervised machine-learning approaches to CF-MS data extremely challenging.

To overcome this limitation, and enable protein interaction network inference in these prokaryotes, we hypothesized that we could train a machine-learning model on known protein complexes across the 206 human and mouse CF-MS experiments. We hypothesized that this would allow the model to learn generic features of protein co-elution across CF-MS experiments in general, and the same model could then be applied directly to predict protein interactions in these four understudied prokaryotes without any further training. Indeed, the results of this experiment demonstrated that this strategy outperformed the classical within-species supervised machine learning paradigm, yielding networks that better separated intra- and inter-complex interactions and which demonstrated a higher degree of network connectivity between proteins annotated with the same GO term on average. We think this is an exciting finding because it opens up new opportunities for CF-MS data analysis in the most understudied organisms, particularly when few protein complexes are known (or can be inferred via ortholog mapping) at all.

We include the results of this experiment in **Fig. 2i** and **Supplementary Fig. 12**, both of which are reproduced below:

Fig. 2 | CFdb prioritizes functional phosphosites and enhances analysis of CF-MS data from non-model organisms.

[...]

i, Separation of intra- and intra-complex interactions in interactome networks reconstructed for four prokaryotes by random forest classifiers trained in cross-validation on species-specific protein complexes (“within-species”) versus on 206 human and mouse CF-MS experiments (“human/mouse”), as quantified by the area under the ROC curve (AUROC).

Supplementary Fig. 12 | Interactome mapping in understudied prokaryotes without a training set of known protein complexes.

a, Receiver operating characteristic (ROC) curves demonstrating separation of intra- and inter-complex interactions by random forest classifiers trained in cross-validation on species-specific protein complexes (“within-species”) versus on 206 human and mouse CF-MS experiments (“human/mouse”).

b, Functional coherence of protein interaction networks for four understudied prokaryotes reconstructed by random forest classifiers trained in cross-validation on species-specific protein complexes (“within-species”) versus on 206 human and mouse CF-MS experiments (“human/mouse”), shown separately for interaction networks at two fixed sizes (top-5,000 versus top-10,000 interactions).

Additionally, we present this experiment in the revised Results section as follows:

We also asked whether CFdb could enable interactome mapping without requiring a training dataset of known protein complexes. To explore this possibility, we focused on four understudied prokaryotes (*Anabaena* sp. 7120, *Cyanothece* sp. ATCC 51142, *Kueneia stuttgartiensis*, and *Synechocystis* sp. PCC 6803) represented in CFdb^{29–32}. Among the known protein complexes in the EcoCyc database³³, just 158 to 190 intra-complex interactions could be mapped to one-to-one orthologs in each of these four species, presenting a challenge to the supervised machine-learning approach that is the dominant paradigm in CF-MS data analysis. We hypothesized that a machine-learning model trained on aggregate patterns of protein complex co-elution across 206 human or mouse CF-MS experiments could enable network inference in these species without requiring a training set of species-specific protein complexes. Consistent with this hypothesis, a random forest classifier trained on human and mouse experiments better separated intra- and inter-complex interactions derived from EcoCyc than supervised machine-learning within each species (**Fig. 2i** and **Supplementary Fig. 12a**). Moreover, protein interaction networks derived from the classifier trained on human and mouse experiments also demonstrated a higher degree of functional coherence than networks derived from within-species machine-learning (**Supplementary Fig. 12b**). These findings indicate that for non-model organisms in which few known protein complexes are available to train a classifier, transfer learning on human and mouse CF-MS datasets can enable *de novo* network inference.

4. What's the difference between supplementary figure 3d and the pink line in supplementary figure 3d? They look identical. If they are the same, suggest combining these two figures. The same as supplementary figure 3 f and g, i and 3j. Supplementary Fig 3e shows that 87% more proteins in 500+ fractions. However, I didn't see such a big increase from Supplementary Fig 3c. How were those values calculated? It's better to provide the original information in the supplementary tables.

The pink line in **Supplementary Fig. 3c** is indeed the same as the black line shown in **Supplementary Fig. 3d**; however, the inset pie charts highlight different aspects of the CFdb resource. Specifically, the panels on the left (**c, f, i**) highlight the difference in protein coverage between versions 1 and 2 of the resource, whereas the center panels (**d, g, j**) highlight the total fraction of the organismal proteome covered in version 2. We agree that there is a degree of redundancy here given that the same data is being presented in both sets of panels, but feel it is still useful for the reader to present the dataset from both standpoints in this supplementary figure.

Regarding the second issue raised in this point, there are 899 mouse proteins quantified in 500+ fractions in version 1, as compared to 1,682 in version 2 (CFdb), an increase of 87%. We have taken your suggestion to provide all of the original data underlying these calculations, which is included in the newly-added **Supplementary Table 2** of the revised manuscript. This table gives the total number of fractions in which each unique protein was quantified across all 32 species in CFdb. We also want to reiterate that we have gone to extensive lengths to make sure the data generated in our study is reusable at multiple different levels; for example, the processed chromatograms for all experiments (which would also allow for this analysis to be reproduced) are available at Zenodo; all of the raw MaxQuant outputs are available from PRIDE; and the code used to generate all the figure panels is available from GitHub (see Data availability and Code availability statements for full details).

5. Supplementary Table 2 only provide the PPIs of human and mouse. Please add PPIs of all species and mark which PPI is also found in version 1 and which is new.

In the original submission, this data was provided via Zenodo. We did attempt to take your suggestion to also provide the PPIs for all species as a supplementary table with the manuscript, but we were unable to

fit all of this data within the 30 MB file size limit of Nature Communications. Instead, we have added the revised Excel file to the Zenodo upload. We updated the link to Zenodo in the manuscript to refer to the revised data deposition (“Data availability” section).

6. Across all fractionation dataset, are there any PPIs changes because of phosphorylation modification?

This is a great example of the kind of biological questions that can now be addressed using the resource of ~750,000 phosphosite quantifications across >21,000 fractions included in CFdb. In fact, using a simple computational approach, we have already identified several examples of protein-protein interactions that appear to be modulated by phosphorylation. One such example is provided in **Fig. 2f**, wherein phosphopeptides for the S25 residue of triosephosphate isomerase are seen to be specific to one particular CF-MS peak—this is consistent with the recently elucidated function of this phosphosite in regulating the homodimerization of this enzyme, which was published while this manuscript was in preparation (Stein et al., *Cancer Discov.* 2023, doi: 10.1158/2159-8290.CD-22-0805). In the revised manuscript, we also show a second example, in which phosphopeptides for the S1347 of nestin are specific to a particular, minor peak whereas most phosphopeptides for this protein are detected in the other, more abundant peak; this hints at a role for phosphorylation of this residue in modulating protein-protein interactions. This figure is reproduced below and included in the revised manuscript as **Supplementary Fig. 7q**:

Supplementary Fig. 7 | Protein phosphorylation across 21,703 CF-MS fractions.

[...]

q, Examples of frequently quantified phosphosites in nestin. Chromatograms show the intensity of pS phosphopeptides (colors) or the parent protein (light grey). One phosphosite (pS1347) is specific to a low-intensity peak in the CF-MS data, whereas other phosphosites are detected exclusively in the highest-intensity peak (note that intensity is shown on a logarithmic scale).

We believe there are many more examples like these to be discovered within CFdb. However, we think that more systematic identification of phosphorylation-dependent interaction (and, importantly, estimating the false discovery rates for these) will require the development of new computational tools, which is a significant undertaking and a conceptual departure from the work presented in this manuscript. Therefore, while we are excited that our work provides a springboard for more extensive characterization of phosphorylation-dependent interactions, this is outside the scope of the present study.

Reviewer #3 (Remarks to the Author):

The manuscript “Mapping protein states and interactions across the tree of life” uses established datasets of CF-MS data to increase coverage of protein interactions and provide more reliable consensus datasets for the broad overview of protein relationships across species, expanding from previous work from a little over 200 datasets to over 400, including (and importantly) expanding to new species. Authors note that many of the interactions that are novel to this dataset over the previous are associated with low abundance or are tissue-specific, which historically has limited protein interaction detection. Interactions found in the new network have a greater overlap with published studies than the original network (fig 5), which suggests the network is improving, and co-localization and co-expression data is promising. Overall, the resource looks to be significant and useful to the scientific community.

Issues:

1. The phosphosite functional relevance prediction via machine learning is interesting as a source of hypothesis-generation or prioritization for further research, however I find certain things less compelling. Authors note that detected phosphosites are more likely to have some source of data for machine learning prediction of function, and I am certain that things that are detectable are statistically more frequently discussed or partially characterized in some way than things that are undetectable, but a better evaluation of this might be to examine whether the specific prediction of function via machine learning matches established or expected function from some other screen or database, such as if cell cycle or DNA damage signaling or stress response kinase targets match the machine learning predictions here assigned.

Thank you for these comments. We are in full agreement with your point that “things that are detectable are statistically more frequently discussed or partially characterized in some way than things that are undetectable,” although we want to emphasize that here, our analysis is comparing the phosphosites detected by CF-MS to those detected in another large meta-analysis of phosphoproteomics data. The phosphosites detected in this meta-analysis were then used as a basis to develop a machine-learning method to predict functional phosphosites on the basis of proteomic, structural, regulatory and evolutionary features. Therefore, the phosphosites with higher functional scores assigned by this approach are not necessarily “more frequently discussed” or even necessarily characterized at all.

With that said, we took your suggestion to examine whether these frequently detected phosphosites match functions from other screens or databases, and specifically to examine their overlap with known kinase targets. We performed a kinase-substrate enrichment analysis, comparing phosphosites detected in five or more CF-MS fractions to those ever detected by CF-MS. We found that these frequently detected phosphosites were enriched for the known targets of several kinases, with the most significant enrichments for CK2A1, CDK1, and CDK2. We also analyzed a number of other structural, signalling, or evolutionary features of frequently detected phosphosites and found that they had a number of unusual features relative to the phosphoproteome average. For instance, phosphosites detected in 5+ CF-MS fractions are more likely to be located in intrinsically disordered regions or phosphorylation ‘hotspots’; more likely to match known protein kinase motifs, and enriched for known substrates of several kinases by kinase-substrate enrichment analysis (KSEA); and more likely to be evolutionarily ancient (e.g., conserved across all tetrapods or bilaterians). We present all of these results in the newly-added **Supplementary Fig. 9**, which is reproduced below:

Supplementary Fig. 9 | Properties of frequently detected phosphosites.

a, Number of MS/MS spectra in which phosphosites were detected in an independent resource of 6,801 phosphoproteomics experiments, in which no samples overlapped with those used in CFdb¹⁶, shown separately for phosphosites detected in 1-4 versus 5 or more CF-MS fractions ($p < 10^{-15}$, Wilcoxon rank-sum test).

b, As in **a**, but showing the number of biological samples in which phosphosites were detected ($p < 10^{-15}$, Wilcoxon rank-sum test).

c, As in **a**, but showing the maximum localization probability with which phosphosites were detected ($p < 10^{-15}$, Wilcoxon rank-sum test).

d, Proportion of phosphosites located within predicted intrinsically disordered regions, shown separately for phosphosites detected in 1-4 versus 5 or more CF-MS fractions ($p = 2.1 \times 10^{-15}$, χ^2 test).

e, As in **d**, but showing phosphosites located within phosphorylation hotspots ($p = 2.0 \times 10^{-8}$, χ^2 test).

f, As in **a**, but showing the maximum kinase position weight motif matrix similarity score for each phosphosite ($p < 10^{-15}$, Wilcoxon rank-sum test).

g, As in **a**, but showing the maximum NetPhorest posterior probability for each phosphosite ($p < 10^{-15}$, Wilcoxon rank-sum test).

h, Kinase-substrate enrichment analysis of phosphosites. Left, number of expected (blue) versus observed (red) substrates of kinases from the PhosphoSitePlus database among phosphosites detected in 5 or more CF-MS fractions, compared to phosphosites ever detected by CF-MS; right, statistical significance of the observed enrichment.

i, As in **d**, but showing the inferred ancestral age of each phosphorylation site ($p = 1.3 \times 10^{-15}$, χ^2 test).

j, Bars, total number of CF-MS fractions collected per major species or taxonomic groups. Text, proportion of fractions per species corresponding to detection in five or more CF-MS fractions.

We also present this data in the revised Results section as follows:

Phosphosites detected in five or more CF-MS fractions had a number of unusual features relative to the phosphoproteome average. In an independent phosphoproteomics dataset, these phosphosites tended to be identified in more samples, by more MS/MS spectra, and with higher localization probabilities, all features characteristic of more reliable identifications (**Supplementary Fig. 9a-c**). They were also more likely to be located in intrinsically disordered regions¹⁹ and in recurrently phosphorylated structural regions (phosphorylation 'hotspots'²⁰), both of which are known to be enriched for functional phosphosites (**Supplementary Fig. 9d-e**). Frequently quantified phosphosites were more likely to match known protein kinase motifs, and kinase-substrate enrichment analysis identified several kinases whose known substrates were enriched among these phosphosites, with the most significant enrichments for CK2A1, CDK1, and CDK2 (**Supplementary Fig. 9f-h**). Last, estimates of phosphosite evolutionary age²¹ suggested that frequently quantified phosphosites were evolutionarily ancient, with human-specific phosphosites depleted and phosphosites conserved across tetrapods or bilaterians enriched among this set (**Supplementary Fig. 9g**).

2. The attempt to use homologous interactions and limited CF-MS screens to improve CF-MS data for honeybees was particularly interesting for the application of this system to non-well characterized species, something that will no doubt become more significant to the research community as proteomics work expands to new organisms. However, it would be more convincing if there was biological data presented beyond correlations with expression or localization, such as example complexes or a promising established structure being identified. Moreover, providing some measures to determine the effectiveness of their established pipeline in other non-model organisms would strengthen the findings.

Thank you for these suggestions. In the revised manuscript, we performed a number of additional analyses to address this comment. First, we analyzed the coverage of CORUM protein complexes in the honey bee interactomes inferred with and without data augmentation using human and mouse datasets from CFdb. As noted in the original manuscript, this analysis is limited because of the relative paucity of CORUM complex proteins that map to one-to-one orthologs in honey bee, but nonetheless, we identified a number of complexes with improved coverage of intra-complex interactions in the network reconstructed with augmentation by human and mouse data from CFdb, such as the 26S proteasome, the LSm2-8 complex, and the 20S methylosome. We include this data in **Supplementary Fig. 11** in the revised manuscript (reproduced at the end of this response).

Second, we searched in the network for promising new interactions that were identified with the data augmentation strategy that we present. We identified an interaction between the metabolic enzyme inositol-3-phosphate synthase and the chaperone/stress response protein calreticulin, which is currently not catalogued in *any* species in BioGRID, but which is supported by convincing co-elution in nine of the eleven CF-MS replicates. Moreover, the same interaction was previously identified by AP-MS in *Drosophila* by the large-scale study of Guruharsha et al. (*Cell* 2011, doi: 10.1016/j.cell.2011.08.047), supporting the presence of an orthologous interaction or 'interolog' in the honey bee.

Third, we took note of your suggestion to demonstrate that CFdb could contribute to the analysis of protein interaction networks for other non-model organisms. We focused our new analysis on four understudied prokaryotes (*Anabaena* sp. 7120, *Cyanothece* sp. ATCC 51142, *Kuenenia stuttgartiensis*, and *Synechocystis* sp. PCC 6803) in CFdb, for which virtually nothing is known about their protein interaction networks. Moreover, attempts to map protein complexes from *E. coli* to their one-to-one orthologs in these species yield extremely small training datasets with just 158 to 190 true positive interactions, which makes standard supervised machine-learning approaches to CF-MS data essentially impossible. To overcome this limitation, and enable protein interaction network inference in these prokaryotes, we hypothesized that we could train a machine-learning model on known protein complexes

across the 206 human and mouse CF-MS experiments, and then apply the same model directly to CF-MS data from these four prokaryotes to predict protein interactions without any further training. Indeed, the results of this experiment demonstrated that this strategy outperformed the classical within-species supervised machine learning paradigm, yielding networks that better separated intra- and inter-complex interactions and which demonstrated a higher degree of network connectivity between proteins annotated with the same GO term on average. We think this is an exciting finding because it opens up new opportunities for CF-MS data analysis in the most understudied organisms, particularly when few protein complexes are known (or can be inferred via ortholog mapping) at all.

The relevant edits to the Results section are reproduced in full below, along with the new panels added to **Fig. 2i** and **Supplementary Fig. 11** and the new **Supplementary Fig. 12**:

To date, the dominant paradigm in the field has been to analyze each newly collected CF-MS dataset in isolation. We asked whether CFdb could augment analyses of smaller-scale CF-MS projects, such as those carried out within individual laboratories. We reasoned that such approaches could be particularly useful for studies of non-model organisms, in which few protein complexes may be known.

As a proof of concept, we carried out a new set of CF-MS experiments in honey bee (*Apis mellifera*). Honey bees are vital pollinators that play central roles in global agriculture^{24,25}. However, the honey bee interactome remains largely unmapped. This gap reflects a number of challenges that are common to the study of non-model organisms, including a lack of established cell culture systems, limited amenability to genetic manipulation, and incomplete proteome annotation. CF-MS is well-suited to overcome these limitations by enabling interactome mapping under physiological conditions within *in vivo* tissues, and without requiring validated antibodies or the introduction of protein tags.

As a first step towards mapping the honey bee interactome, we profiled the honey bee midgut by CF-MS. The midgut was selected as the primary site of infection for a prevalent honey bee pathogen, *Vairimorpha* (*Nosema*) *cerana*, that has been implicated in the collapse of honey bee colonies. Ten CF-MS experiments were performed in which 40 fractions were collected and mass spectrometry data was acquired with data-independent acquisition (DIA), while an eleventh experiment was analyzed using data-dependent acquisition (DDA). An average of 2,092 proteins were quantified per replicate, yielding a total of 319,105 protein quantifications across all 440 fractions.

We then sought to infer protein interaction networks for the honey bee midgut from this dataset. However, of the 5,163 human proteins within the CORUM database¹⁰, only 1,200 could be mapped to a bee ortholog, limiting the amount of training data available for network inference. We therefore devised a **data augmentation** strategy that leveraged CFdb to augment our bee data with labelled protein pairs that were sampled randomly from 206 human or mouse CF-MS experiments (**Supplementary Fig. 11a**). This strategy increased the size of the honey bee CF-MS interactome by 64.7%, while simultaneously improving its functional coherence (**Fig. 2h-i**, **Supplementary Fig. 11b-c** and **Supplementary Table 5**). Moreover, interacting proteins in either network showed comparable patterns of coexpression across a large honey bee proteomic dataset²⁶, and their fly orthologs displayed comparable patterns of co-elution in an independent CF-MS dataset²⁷ (**Supplementary Fig. 11d-e**). Data augmentation enabled improved coverage of several CORUM protein complexes with one-to-one orthologs in honey bee, such as the 26S proteasome, the LSM2-8 complex, and the 20S methylosome (**Supplementary Fig. 11f**). Beyond better coverage of known complexes, data augmentation also enabled the identification of novel interactions. For instance, we identified an interaction between calreticulin and inositol-3-phosphate synthase. This interaction was not annotated in any species in the BioGRID database, but these proteins were previously found to co-purify in a large AP-MS study of the fly interactome²⁸, supporting the existence of an orthologous interaction in honey bee (**Supplementary Fig. 11g-h**).

We also asked whether CFdb could enable interactome mapping without requiring a training dataset of known protein complexes. To explore this possibility, we focused on four understudied prokaryotes (*Anabaena* sp. 7120, *Cyanothece* sp. ATCC 51142, *Kueneenia stuttgartiensis*, and *Synechocystis* sp. PCC 6803) represented in CFdb²⁹⁻³². Among the known protein complexes in the EcoCyc database³³, just 158 to 190 intra-complex interactions could be mapped to one-to-one orthologs in each of these four species, presenting a challenge to the supervised machine-learning approach that is the dominant paradigm in CF-MS

data analysis. We hypothesized that a machine-learning model trained on aggregate patterns of protein complex co-elution across 206 human or mouse CF-MS experiments could enable network inference in these species without requiring a training set of species-specific protein complexes. Consistent with this hypothesis, a random forest classifier trained on human and mouse experiments better separated intra- and inter-complex interactions derived from EcoCyc than supervised machine-learning within each species (**Fig. 2i** and **Supplementary Fig. 12a**). Moreover, protein interaction networks derived from the classifier trained on human and mouse experiments also demonstrated a higher degree of functional coherence than networks derived from within-species machine-learning (**Supplementary Fig. 12b**). These findings indicate that for non-model organisms in which few known protein complexes are available to train a classifier, learning from human and mouse CF-MS datasets can enable *de novo* network inference.

Fig. 2 | CFdb prioritizes functional phosphosites and enhances analysis of CF-MS data from non-model organisms.

[...]

i, Separation of intra- and intra-complex interactions in interactome networks reconstructed for four prokaryotes by random forest classifiers trained in cross-validation on species-specific protein complexes (“within-species”) versus on 206 human and mouse CF-MS experiments (“human/mouse”), as quantified by the area under the ROC curve (AUROC).

Supplementary Fig. 11 | Mapping the honey bee interactome with data augmentation using CFdb.

[...]

f, Examples of CORUM protein complexes with one-to-one orthologs in honey bee with intra-complex interactions better resolved by network inference after data augmentation.

g, Precision with which the putative interaction between calreticulin and inositol-3-phosphate synthase was recovered the honey bee CF-MS interactome mapped with or without data augmentation.
h, Elution profiles of calreticulin and inositol-3-phosphate synthase in nine CF-MS replicates in which both proteins were quantified.

Supplementary Fig. 12 | Interactome mapping in understudied prokaryotes without a training set of known protein complexes.

a, Receiver operating characteristic (ROC) curves demonstrating separation of intra- and inter-complex interactions by random forest classifiers trained in cross-validation on species-specific protein complexes (“within-species”) versus on 206 human and mouse CF-MS experiments (“human/mouse”).

b, Functional coherence of protein interaction networks for four understudied prokaryotes reconstructed by random forest classifiers trained in cross-validation on species-specific protein complexes (“within-species”) versus on 206 human and mouse CF-MS experiments (“human/mouse”), shown separately for interaction networks at two fixed sizes (top-5,000 versus top-10,000 interactions).

3. The authors used high throughput data from 2014, it would be nice to discuss how the potential biases introduced by the following factors (in the studies) were mitigated or considered during their analysis:

- Quality of data because of variation in experimental protocols between studies.
- Biological systems are dynamic and may change over time and interactions might be context dependent or transient.
- Y2H and AP-MS are two different methods for interaction detection having their own merits and demerits – Y2H might have missed interactions in the context of human cells and AP-MS might have false positive interactions due to purification processes.

We agree that these are all important considerations that could have affected the overlap between high-throughput screens, most of which have been extensively discussed in the literature as potential reasons for the limited overlap (e.g. by references 4 and 5, both published in 2002). We revised the paragraph in question to raise these considerations as shown below:

Cellular processes arise from the dynamic organization of proteins in networks of physical interactions. Significant resources have been devoted to mapping the protein interaction networks of humans and model organisms¹. These networks are widely used for tasks such as protein function prediction, disease gene prioritization, or interpretation of transcriptomic and proteomic datasets^{2,3}. However, questions about the reproducibility of these networks have persisted. Limited overlap between screens performed in different laboratories was noted soon after the first maps of the yeast interactome emerged^{4,5}. Two decades later, large-scale efforts have produced systematic maps of the human interactome that display relatively little overlap with one another, with a mean Jaccard index of just 0.062 between any pair of networks (**Supplementary Fig. 1**). This lack of overlap has been variously attributed to differences in the types of interactions detected by each assay and the proteins targeted by individual screens, variation in experimental protocols or the depth of protein identification, the presence of context-specific interactions, or false-positives and false-negatives in the resulting interactome maps.

Moreover, in response to a comment by reviewer 2, we now show the overlap between the consensus CF-MS interactome and each of these high-throughput screens, in **Supplementary Fig. 5k**. This analysis, we feel, helps provide further context for the Jaccard index between pairs of past screens.

Finally, we fully agree that many interactions are context-dependent (our own past work has explored this concept extensively) and we provide additional discussion of this point in response to point #9 below.

4. I wonder why in the original meta-analysis (“version 1”) no phosphosite was detected. All the sites in (Fig 1b last graph) are detected in the version 2 analysis (in pink).

This is because in our original meta-analysis, we had not carried out a search for phosphopeptides. Identifying phosphorylation events in the 206 datasets included in our original analysis therefore required us to re-search all 206 datasets for phosphopeptides with MaxQuant, a significant undertaking that we think adds to the value of the present resource. We revised the figure legend to clarify this point:

Fig. 1 | A harmonized resource of CF-MS data charts protein abundance and interactions across the tree of life.

[...]

b, Expansions to the scope and coverage of CF-MS data in CFdb (“version 2”), as compared to our original meta-analysis (“version 1”). Phosphosite quantifications are assigned exclusively to version 2 because CF-MS datasets were not searched for phosphopeptides in our original meta-analysis.

5. The improvement (Fig. 1c-e) after doubling the resource size might still be biased – towards certain species and/or proteins (less abundant or less studied).

This is certainly true. With respect to species biases, our meta-analysis is constrained by the experiments that other laboratories have elected to carry out and the implication is that the resource focuses primarily on human, *Arabidopsis*, and mouse, as shown in **Supplementary Fig. 2c-d**. This is exciting in the sense that the vast amounts of data available for these species affords great power to identify reproducible interactions, a point on which we elaborate further below. But it is also true that there is tremendous opportunity to apply CF-MS more broadly across the phylogenetic tree (one could argue that CF-MS might be most useful to map protein interactions in less well-studied organisms, where techniques like AP-MS or Y2H are less feasible). With respect to protein abundance, we agree that this is one of the most important biases of CF-MS, and while we show that the expansion of the human, mouse, and *Arabidopsis* datasets has allowed us to greatly mitigate the bias towards detection of highly-abundant proteins by

detecting many more low-abundance proteins, these still only tend to be quantified in a relatively small number of fractions.

To address these points, we added a new paragraph to the Discussion section of the revised manuscript, reproduced below:

Our meta-analytical approach also meant that we were limited both by the biological scope of the CF-MS datasets that have been deposited to public repositories, as well as the technical limitations of CF-MS itself. For example, whereas CFdb encompasses large resources of uniformly processed CF-MS data from human, *Arabidopsis*, and mouse, there are clearly still opportunities to apply CF-MS more broadly in less-studied organisms (**Supplementary Fig. 2c-d**). Similarly, while the expansion of CF-MS data in these species allowed us to detect many low-abundance proteins that were not observed in our original resource, this does not negate the more general bias of CF-MS towards highly abundant proteins, which tend to be quantified in the greatest number of fractions (**Supplementary Fig. 4b-c**).

6. Findings in this paper such as interactions, and functional relevance of phosphosites (Fig. 2c and d) should be confirmed with established experimental methods (co-immunoprecipitations, functional assays, etc.) to ensure accuracy and reliability.

We fully agree that, although our analyses provide strong support for the quality of the inferred interaction network at the systems level, any particular individual interaction should be regarded as putative until confirmed experimentally using an orthogonal technique. With that said, we believe that ‘validating’ a few interactions out of hundreds of thousands of predicted interactions does very little to increase the overall confidence in the dataset, since it is such a miniscule fraction of the total. (Of note, this kind of experimental validation can be exceedingly challenging in biological systems not amenable to conventional approaches such as affinity purification, e.g., honey bee.) Instead, we believe that a far better validation of the data is through extensive comparisons to other functional genomics datasets, as we have done extensively in the manuscript. We revised the manuscript to comment on these points, at the end of the paragraph below:

Discussion

The increasing uptake of CF-MS has led to the deposition of hundreds of datasets in public proteomic repositories. However, these datasets are generally collected and analyzed in isolation by individual laboratories. Here, we explored the possibility of aggregating biologically and technically heterogeneous CF-MS data at the repository scale. We re-analyzed 21,703 fractions from 411 CF-MS experiments using a uniform computational pipeline that standardized protein identification, quantification, and quality control. This expanded resource incorporated data from eight new species and substantially expanded the proteome coverage of humans and model organisms by CF-MS. It also dramatically improved our ability to infer protein interaction networks through meta-analysis of all available CF-MS data for any given species. For example, through meta-analysis of 166 CF-MS experiments, we have produced a map of the human interactome that multiple lines of evidence suggest is among the highest-quality interactome maps currently in existence. Similarly, we present a high-quality map of the mouse interactome derived from meta-analysis of 40 CF-MS experiments, as well as CF-MS interactomes for major model organisms not covered in our original resource such as yeast or *E. coli*. These networks include interactions for hundreds of low-abundance, tissue-specific, and/or understudied proteins not captured in our original resource. We carried out extensive comparisons to other functional genomics datasets that substantiate the quality of the inferred networks at a systems level. However, we caution users of this resource that any particular interaction should in isolation be regarded as putative until confirmed experimentally using an orthogonal technique.

7. The article lacks an in-depth discussion of potential technical limitations associated with CF-MS and the impact of data preprocessing, quality control, and potential biases on the resulting interaction networks.

We certainly agree that CF-MS has a number of technical limitations. As noted above, and shown in **Supplementary Fig. 2b-c**, CF-MS is biased towards more abundant proteins, which tend to be quantified in a greater number of fractions and experiments. CF-MS will also expectantly be biased towards interactions in large, stable protein complexes and depleted for transient interactions, although this will depend to some extent on sample preparation. We have added new text to the Discussion to address these points, as shown above (point #5) as well as below (point #8).

With respect to data preprocessing and quality control, we feel the use of a uniform pipeline to re-process and QC raw data from all 411 studies is a major strength of our study. We felt that the differences in data processing/QC from one study to another (e.g. protein quantification, the protein sequence databases searched, or filtering low-quality chromatograms) would unnecessarily hinder our ability to identify reproducible interactions—in other words, it would introduce noise without any commensurate increase in signal. This was the reason we undertook the substantial effort of re-processing all 411 experiments from scratch using a consistent preprocessing pipeline, including protein detection, quantification, and quality control. We now emphasize this in the Discussion—the relevant text is reproduced below in response to point #9, which raises a similar point.

8. The article focuses predominantly on CF-MS and its advantages, but it could benefit from discussing how the presented approaches and insights could complement or be integrated with other proteomic techniques.

We agree that there are exciting opportunities to integrate many different types of proteomic datasets together in order to derive more accurate and comprehensive interaction networks. This kind of integration is perhaps so far best exemplified by the hu.MAP resources (Drew et al., *Mol. Syst. Biol.* 2017 and 2021), which initially integrated data from one CF-MS study (Wan et al., *Nature* 2015) and two AP-MS studies (Huttlin et al., *Cell* 2015 and Hein et al., *Cell* 2015) using a machine-learning pipeline very similar to that used in our study. The authors subsequently developed an updated version of this resource in which they integrated additional AP-MS datasets, two proximity labeling datasets, and an RNA hairpin pulldown dataset to more accurately identify interacting protein pairs. These efforts therefore benefited from integrating a much broader range of proteomic datasets than we have considered here, but they made use of data from relatively few (albeit unusually large-scale) studies.

We think that the availability of uniformly processed data from hundreds of CF-MS experiments in CFdb provides a springboard for the development of even more comprehensive and accurate proteome resources. These might draw on not just AP-MS and CF-MS but also proximity labelling, thermal proximity co-aggregation (e.g. Jarzab et al., *Nat. Methods* 2020), protein co-abundance (e.g. Kustatscher et al., *Nat. Biotechnol.* 2019), or structure-based computational inferences (e.g. O'Reilly et al., *Mol. Syst. Biol.* 2023) to identify protein interactions. We have revised the Discussion section to comment on these kinds of opportunities, and have made a particular effort to discuss these in the context of the limitations of CF-MS as noted in response to the preceding point. The new paragraph is reproduced below:

Beyond the meta-analyses of CF-MS data described here, CFdb also provides a springboard for the development of even more comprehensive and accurate interactome resources by integrating CF-MS data with data from other proteomic techniques. Efforts to this end^{22,23} have demonstrated the feasibility of integrating CF-MS data with AP-MS and proximity labelling datasets to develop integrated interaction

networks, using a machine-learning approach similar to that employed here. Future efforts in this vein might additionally draw on thermal protein co-aggregation²⁴, protein co-abundance^{25,26}, or structure-based computational inferences^{27,28} to further increase the scope and accuracy of network inference. Notably, combining data from multiple orthogonal techniques could at least partially mitigate some of the inherent biases of CF-MS, such as its preference for stable macromolecular complexes over transient interactions.

9. However, comparing and integrating CF-MS data from different studies can be challenging due to variations in experimental protocols, data acquisition methods, and data processing approaches. This should be specified in the text, as well as how they overcome these challenges. Moreover, the human or mouse CF-MS data are most likely retrieved from different cell lines or tissues. Considering the proteome profile differences (i.e., proteins are expressed at different levels depending on the human cell model studied) for each cell line/tissue, a global human interactome could be limiting. Since for each cell-line/tissue the protein levels are different, protein-protein interactions vary as well. This should be addressed, specifying the possible differences in the interactome depending on the tissue analyzed.

In our reading, this comment raises two distinct issues. Perhaps the most important one is regarding the biological heterogeneity of the underlying data, i.e., that our approach uses CF-MS data from so many different cell lines or tissues in each species. We certainly agree that the concept of a global human interactome has limitations. In fact, we have applied CF-MS extensively to understand how the interactome is 'rewired' across physiological contexts (e.g. Kristensen et al., *Nat. Methods* 2012; Scott et al., *Mol. Syst. Biol.* 2017; Kerr et al., *Genome Biol.* 2020; Skinider et al., *Cell* 2021). That being said, the goal of our meta-analytical approach in this study was to use the vast quantities of CF-MS data in public repositories to discover the most robust interactions—those supported by data from dozens or even hundreds of different experiments. In other words, we sought to identify pairs of proteins that consistently had correlated interaction profiles across many different experimental protocols and data acquisition methods. Our expectation was that these pairs would be the most likely to be *bona fide* interactors, and we could separate signal from noise to accurately identify these pairs by drawing on dozens or hundreds of experiments. In the revised Discussion, we have clarified that our goal here was to identify the most robust and global interactions and acknowledge that this approach will necessarily exclude many context-specific interactions that might otherwise be detected by focusing on individual biological systems.

Our meta-analytical approach has both strengths and limitations. Our goal in this study was to leverage the vast quantities of CF-MS data that have been deposited to public repositories in order to identify protein interactions supported by evidence from many biologically and technically heterogeneous experiments. In other words, we sought to identify protein pairs that consistently demonstrated co-elution across different cell lines or tissues, experimental protocols, and mass spectrometric methods, with the expectation that these pairs would be the most likely to represent *bona fide* interactions. We reasoned that drawing on dozens or hundreds of experiments in any given species would allow us to separate signal from noise to accurately identify these pairs. To minimize unnecessary variation in data preprocessing or quality control, and further increase our ability to separate signal from noise, we re-processed the raw data from all 411 CF-MS experiments through a uniform pipeline, which is a significant strength of this resource. On the other hand, the goal of identifying robust, global interactions is at odds with the application of CF-MS to identify context-specific interactions that might only be detected when focusing on particular biological systems, as we and others have explored^{6,9,22–26}. These context-specific interactions will expectantly require more focused analyses of smaller (but biologically and technically homogenous) datasets to detect.

Separately, with respect to the *technical* heterogeneity of the underlying data, we are in full agreement that differences in data processing approaches can complicate the comparison and integration of CF-MS datasets. This was exactly our motivation to systematically reprocess all 411 datasets in this study using a uniform computational pipeline, i.e., with the same software (MaxQuant version 1.6.5.0) used for protein

identification and quantification, the same protein sequence databases, and the same approaches to filtering low-quality chromatograms. In the paragraph above, we also comment on this point as being a strength of our study.

Minor questions:

1. I am unsure as to the value of 'instrument time' as a metric of the meta-analysis.

We agree that 'instrument time' is not a metric of the dataset quality in and of itself. However, we do think the total instrument time required to analyze all >21,000 fractions helps express in readily understandable terms just how much data has been incorporated in the meta-analysis, and why meta-analysis is a valuable approach to develop resources that could not realistically be collected within a single laboratory. We revised the sentence in question to clarify these points:

[...] The updated resource, which we named CFdb, now comprises 20.1 million measurements of protein abundance derived from 128.7 million sequenced peptides across 21,703 fractions (**Fig. 1a-b**). **Proteomic analysis of all 21,703 fractions required a total of 43.2 months of uninterrupted instrument time, emphasizing the value of meta-analysis to assemble a resource whose scale would make it impractical to acquire within a single laboratory.**

2. The significant increase in tyrosine residues as phosphorylation sites could do with more comment, as this is a departure from expectations (as authors note). Why are so many more tyrosine sites now being identified? This would only require a minor hypothesis.

Thank you for this comment. To clarify, the hypothesis that we developed from this observation is that tyrosine phosphosites are overrepresented within protein complexes. This hypothesis led us to carry out a follow-up experiment in which we analyzed a large meta-analysis of phosphoproteomic data (Ochoa et al., *Nat. Biotechnol.* 2020), and we found that phosphotyrosines were enriched among protein complexes in this dataset as well. We have clarified our hypothesis and the motivation for carrying out this second enrichment analysis in the revised manuscript. In addition, and in line with your major point #1 above, we also tested whether the enrichment for phosphotyrosines among protein complex-containing proteins could be confounded by differences in protein abundance, but found that there was still a significant enrichment even when controlling for protein abundance. We revised this section of the manuscript as follows:

Interestingly, whereas radioisotope¹³ and phosphoproteomic¹⁴ data suggest that phosphotyrosines (pY) account for less than 1% of the phosphoproteome, we observed a substantially greater proportion of pY sites in CF-MS data (**Fig. 2b** and **Supplementary Fig. 8a-c**). **Tyrosine residues accounted for 15.1% of the phosphosites detected by CF-MS, similar to the proportion in the curated PhosphoSitePlus database¹⁵ (16.5%) but greater than that observed in the largest meta-analysis of phospho-enrichment proteomic datasets to date (5.1%)¹⁶. Given that most CF-MS protocols enrich for protein complexes by design, this observation raised the possibility that tyrosine phosphosites might be overrepresented within protein complexes. In agreement with this possibility, pY-containing proteins were also enriched among protein complexes in a large-scale meta-analysis of phosphoproteomic datasets¹⁶ (**Supplementary Fig. 8d**), and this enrichment remained statistically significant when controlling for protein abundance ($p = 9.4 \times 10^{-4}$, logistic regression).**

3. Missing caption of the panels f and g in the supplementary Fig. 4.

Fixed, thank you:

Supplementary Fig. 4 | Properties of proteins detected by CF-MS.

[...]

f, Saturation analysis showing the number of human, mouse, and *Arabidopsis* proteins ever detected by CF-MS when sampling experiments in random order from CFdb.

g, Projection of future increases in the number of human proteins ever detected by CF-MS after doubling the number of human CF-MS experiments in CFdb, based on data from **f**.

REVIEWERS' COMMENTS

Reviewer #1 (Remarks to the Author):

The authors have addressed my major comments and considerably expanded on the results and methods. The paper now reads much better than the compact format initially submitted. I appreciate the effort to rephrase a number of statements to take a nuanced stance on the contribution of the data and to clearly define the performance metrics. I am now convinced that the manuscript is ready for publication.

Reviewer #2 (Remarks to the Author):

All my concerns have been addressed in the revisions.

Reviewer #3 (Remarks to the Author):

I am satisfied with the revision, they answer or justify things pretty extensively. A lot of sections seem greatly expanded, and I like the increased discussion of where the phosphosites are found. My only complaint would be the hypothesis that pY sites happen more frequently in complexes is not clear to me, and this comes up twice. Whether they mean pY are part of complex formation/regulation or otherwise part of what makes up a complex (a strong hypothesis that would need more justification) or pY is, for unknown reasons, common in large assemblies and rare in proteins that act independently (I can't think of why that would be, and most proteins act as part of complexes/assemblies, but I think this is what they mean). However this was a minor issue, and I did say it only required a minor hypothesis, so I wouldn't be against publication over it. Overall, this is an impressive work and the resources generated will be more valuable to the scientific community.

Response to reviewers

Reviewer #1 (Remarks to the Author):

The authors have addressed my major comments and considerably expanded on the results and methods. The paper now reads much better than the compact format initially submitted. I appreciate the effort to rephrase a number of statements to take a nuanced stance on the contribution of the data and to clearly define the performance metrics. I am now convinced that the manuscript is ready for publication.

Thank you for your constructive feedback on our work!

Reviewer #2 (Remarks to the Author):

All my concerns have been addressed in the revisions.

Thank you for your constructive feedback on our work!

Reviewer #3 (Remarks to the Author):

I am satisfied with the revision, they answer or justify things pretty extensively. A lot of sections seem greatly expanded, and I like the increased discussion of where the phosphosites are found. My only complaint would be the hypothesis that pY sites happen more frequently in complexes is not clear to me, and this comes up twice. Whether they mean pY are part of complex formation/regulation or otherwise part of what makes up a complex (a strong hypothesis that would need more justification) or pY is, for unknown reasons, common in large assemblies and rare in proteins that act independently (I can't think of why that would be, and most proteins act as part of complexes/assemblies, but I think this is what they mean). However this was a minor issue, and I did say it only required a minor hypothesis, so I wouldn't be against publication over it. Overall, this is an impressive work and the resources generated will be more valuable to the scientific community.

Thank you for your constructive feedback on our work!